# Depolarization of multidomain ferroelectric materials

Dong Zhao[1,2], Thomas Lenz[1,3], Gerwin H. Gelinck[4,5], Pim Groen[4,6], Dragan Damjanovic [7], Dago M. de Leeuw[1,6] & Ilias Katsouras[4]

Depolarization in ferroelectric materials has been studied since the 1970s, albeit quasi-statically. The dynamics are described by the empirical Merz law, which gives the polarization switching time as a function of electric field, normalized to the so-called activation field. The Merz law has been used for decades; its origin as domain-wall depinning has recently been corroborated by molecular dynamics simulations. Here we experimentally investigate domain-wall depinning by measuring the dynamics of depolarization. We find that the boundary between thermodynamically stable and depolarizing regimes can be described by a single constant, $P_r/\varepsilon_0\varepsilon_{ferro}E_c$. Among different multidomain ferroelectric materials the values of coercive field, $E_c$, dielectric constant, $\varepsilon_{ferro}$, and remanent polarization, $P_r$, vary by orders of magnitude; the value for $P_r/\varepsilon_0\varepsilon_{ferro}E_c$ however is comparable, about 15. Using this extracted universal value, we show that the depolarization field is similar to the activation field, which corresponds to the transition from creep to domain-wall flow.

[1] Max-Planck Institute for Polymer Research, Ackermannweg 10, 55128 Mainz, Germany. [2] Max-Planck Institute for Polymer Solid-State Research, Heisenbergstr. 1, 70569 Stuttgart, Germany. [3] Graduate School Materials Science in Mainz, Staudingerweg 9, 55128 Mainz, Germany. [4] Holst Centre, High Tech Campus 31, 5656 AE Eindhoven, The Netherlands. [5] Eindhoven University of Technology, 5600 MB Eindhoven, The Netherlands. [6] Faculty of Aerospace Engineering, Delft University of Technology, Kluyverweg 1, 2629 HS Delft, The Netherlands. [7] Group for Ferroelectrics and Functional Oxides, Swiss Federal Institute of Technology—EPFL, 1015 Lausanne, Switzerland. Correspondence and requests for materials should be addressed to D.Z. (email: d.zhao@fkf.mpg.de) or to I.K. (email: ilias.katsouras@tno.nl)

The existence of a depolarization field in polarized ferroelectric materials was experimentally demonstrated in the 1970s[1,2]. In the pioneering work of Wurfel et al.[2], a triglycine sulfate ferroelectric film was sandwiched between a metal electrode and a p-type silicon counter electrode. The ferroelectric material could be poled when the Si-semiconductor was strongly illuminated. The photo-generated charge carriers then could stabilize both the positive and negative ferroelectric polarizations. However, after switching off the light, there is no accumulation of majority carriers to provide sufficient charge compensation, hence only half of the polarization loop was observed. Without compensating charges, the ferroelectric material cannot maintain the remanent polarization due to the presence of a large depolarization field.

A polarized ferroelectric capacitor exhibits a depolarization field whose magnitude can be derived from classical electrostatics within the boundary conditions of a thin film and is given by[3]

$$E_{dep} = -\frac{P}{\varepsilon_0 \varepsilon_{ferro}} \qquad (1)$$

where $P$ is the ferroelectric polarization averaged over the film area, $\varepsilon_0$ and $\varepsilon_{ferro}$ are the vacuum permittivity and static dielectric constant of the material, respectively. As a typical example, for the ferroelectric random copolymer poly(vinylidenefluoride–trifluoroethylene) [P(VDF–TrFE)], the depolarization field can be estimated as 1 GV/m using a remanent polarization of 7 μC/cm$^2$ and a static dielectric constant of 10. This field is an order of magnitude higher than the experimental coercive field, $E_c$, of about 50 MV/m. The polarization is expected to be unstable, but can be stabilized in a ferroelectric capacitor as the metallic electrodes provide free charges that fully compensate the depolarization field, yielding a zero internal electric field inside the ferroelectric material.

When the depolarization field is not fully compensated, the remanent polarization is suppressed. In Supplementary Note 2, we argue that, for our experiments, extrinsic effects, such as occurrence of a capacitive dead layer between the electrodes and the ferroelectric material, or finite screening length in metallic electrodes can be disregarded. Depolarization can then be modeled using an equivalent circuit comprising a linear capacitor in series with a ferroelectric capacitor[4–6]. This circuit yields a relation between depolarization field and suppressed remanent polarization, and has been verified by quasi-static hysteresis loop measurements[3], where the electric field is gradually changed. However, when a high applied electric field is abruptly switched off, the final polarization state is not a priori known; as the polarization is a highly non-linear function of the electric field, one might even expect macroscopic polarization reversal as an overshoot effect.

Polarization reversal in a defect-free single-crystal originates from a coherent, collective rotation of dipoles[7], which is termed intrinsic switching. However, in commonly used ferroelectric materials the presence of defects lowers the barrier for polarization reversal. Switching is then extrinsic and occurs by nucleation and anisotropic growth of individual domains. The domain-wall motion is typically described by a creep velocity[8]. The reciprocal domain-wall velocity is proportional to the switching time[9], which follows the empirical Merz law[10]:

$$t_0 = t_\infty \exp\left(\frac{E_{act}}{E}\right). \qquad (2)$$

where $t_\infty$ is the switching time at infinite applied electric field and $E_{act}$ is the temperature-dependent activation field[11], which is related to the energy required to move the domain walls.

The Merz law has been used empirically for decades. The exact origin of the activation field remains elusive but has recently been related by first-principles-based molecular dynamics simulations to domain-wall depinning.

Here we study experimentally the dynamics of depolarization. To this end, we use the model system of a discrete ferroelectric capacitor in series with a linear capacitor to tune the compensation of the depolarization field[3]. We used two classic multidomain ferroelectric materials, viz. polycrystalline P(VDF–TrFE) and ceramic Pb(Zr,Ti)O$_3$ (PZT), where polarization switching is mediated by nucleation and growth of domains. After applying a voltage pulse the electric displacement is measured as a function of time, from which the internal electric field is derived. The transients are quantitatively described by a generalized Kolmogorov–Avrami–Ishibashi (KAI) formalism[12–14].

We use the transients to construct a depolarization diagram from the normalized displacement as a function of the ratio of the capacitances, which shows depolarizing and thermodynamically stable regimes. The boundary between these regimes defines a unique constant, comprising coercive field, dielectric constant, and saturated polarization, $P_r/\varepsilon_0\varepsilon_{ferro}E_c$, of about 15. This experimentally derived constant is similar for PZT and P(VDF–TrFE). Remarkably, the values of $P_r/\varepsilon_0\varepsilon_{ferro}E_c$ for a large number of other multidomain ferroelectric materials are comparable as well, although the values of coercive field, dielectric constant, and polarization vary by orders of magnitude, which indicates a universal character of the extracted constant.

We note, for completeness, that a similar relation between the intrinsic, thermodynamic coercive field and ferroelectric polarization has theoretically been derived in the seminal work of Tagantsev et al.[15] using the Ginzburg–Landau–Devonshire formalism. Despite the apparent similarity, the physical mechanism of intrinsic switching is completely different from the nucleation and growth studied here for extrinsic switching.

Here we show that our dynamic measurements and the experimental extraction of the relation from the depolarization diagram allows us to draw a link between the depolarization field and the activation field for domain-wall depinning in disordered ferroelectric systems. Using the experimentally extracted universal value of $P_r/\varepsilon_0\varepsilon_{ferro}E_c$ of about 15, we show that the depolarization field is similar to the activation field, $E_{dep} \sim E_{act}$, which corresponds to the transition from creep to domain-wall flow. We argue that this causality naturally holds, when domain-wall depinning, at super-switching fields above the $E_c$, originates from switching of polarized regions near strong pinning sites.

## Results

**Suppression of the remanent polarization.** We first set the framework for studying depolarization, by using quasi-static measurements. The displacement versus applied voltage, $D$-$V_{app}$, hysteresis loops of ferroelectric-only capacitors comprising PZT or P(VDF–TrFE) are presented in Fig. 1a and d, respectively. The fabrication of the capacitors is described in the "Methods" section. Values extracted for the coercive field amounted to 1 and 50 MV/m, and values extracted for the remanent polarization amounted to 38 and 7 μC/cm$^2$, respectively, all in good agreement with literature[16,17].

To tune the compensation of the depolarization field, we connect a linear capacitor, $C_{ser}$, in series to the ferroelectric capacitor, $C_{ferro}$. The hysteresis loops of ferroelectric capacitors based on PZT and P(VDF–TrFE) in series with different linear capacitors are presented in Fig. 1b and e, respectively. At high bias the displacement is the same as for the ferroelectric-only capacitor. For larger ratios of $C_{ferro}/C_{ser}$, a higher applied voltage is needed to fully polarize the ferroelectric capacitor. All loops have an identical apparent coercive voltage, independent of the ratio $C_{ferro}/C_{ser}$, as at zero displacement there are no net free

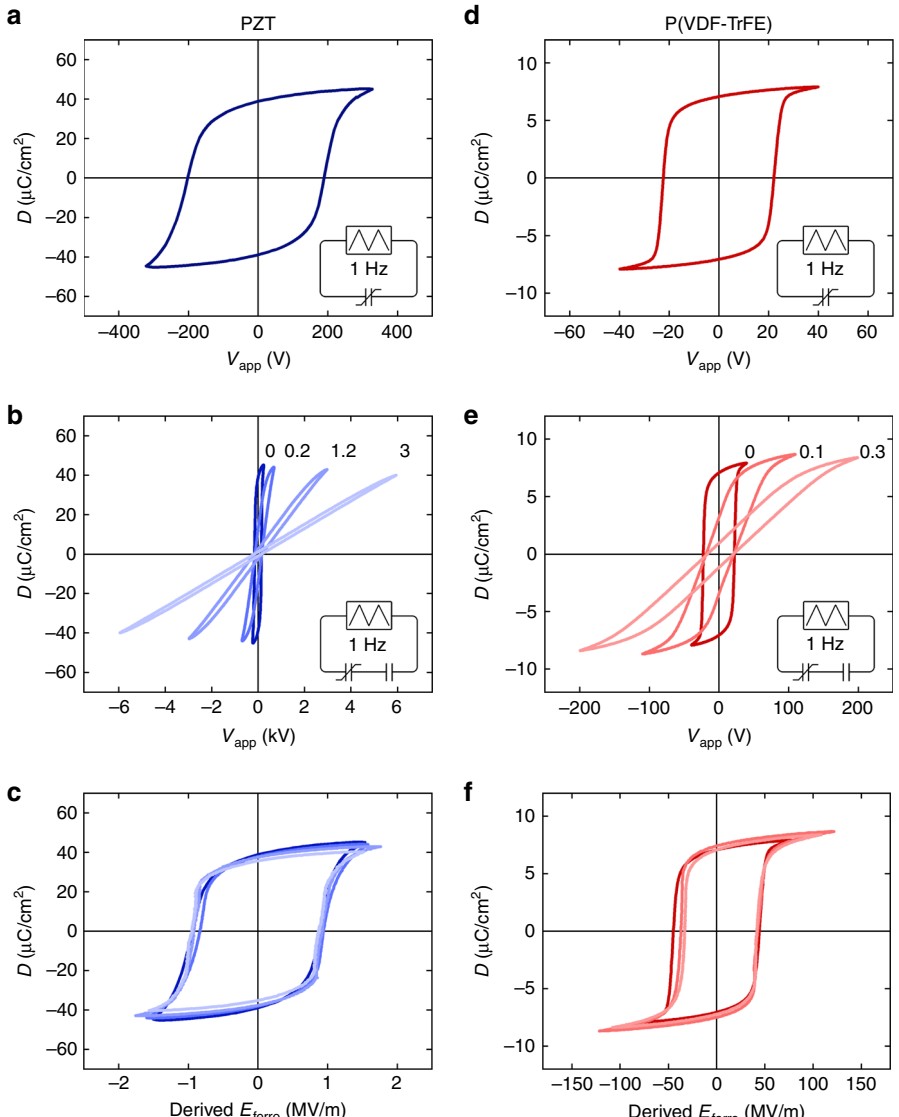

**Fig. 1** Quasi-static hysteresis loops. **a**, **d** Electric displacement vs. applied voltage measured quasi-statically at 1 Hz on a ceramic Pb(Zr, Ti)O$_3$ (PZT) and poly(vinylidenefluoride–trifluoroethylene) [P(VDF–TrFE)] ferroelectric-only capacitor, respectively. The insets schematically show the measurement circuit. **b**, **e** Electric displacement vs. applied voltage measured at 1 Hz on a PZT and P(VDF–TrFE) ferroelectric capacitor in series with different linear capacitors. The ratio of capacitances varies from 0 to 3 for PZT and from 0 to 0.3 for P(VDF–TrFE), and is indicated. At high bias the displacement is the same as for the ferroelectric-only capacitor. For larger ratios of $C_{\mathrm{ferro}}/C_{\mathrm{ser}}$, a higher applied voltage is needed to fully polarize the ferroelectric capacitor. **c**, **f** Reconstructed loops as a function of the electric field inside the ferroelectric material, $E_{\mathrm{ferro}}$, which is derived by classical electrostatics as described in the section "Methods". The reconstructed loops for various $C_{\mathrm{ferro}}/C_{\mathrm{ser}}$ ratios are similar to that of the ferroelectric-only capacitor

charges in the electrodes and hence the equivalent circuit is a ferroelectric-only capacitor. As a consequence of their similar values of coercive field and displacement at high bias, the hysteresis loops get tilted.

To demonstrate that the properties of the ferroelectric material are not changed by adding the serial capacitor, we reconstructed the hysteresis loops as a function of the electric field inside the ferroelectric material, $E_{\mathrm{ferro}}$, which is derived by classical electrostatics in the section "Methods" as[18]

$$E_{\mathrm{ferro}} = \frac{V_{\mathrm{app}}}{d} - \frac{D}{\varepsilon_0 \varepsilon_{\mathrm{ferro}}} \cdot \frac{C_{\mathrm{ferro}}}{C_{\mathrm{ser}}} \qquad (3)$$

where $V_{\mathrm{app}}$ is the applied voltage and $d$ is the thickness of the ferroelectric layer. Note that due to the incomplete compensation at $V_{\mathrm{app}} = 0$ the net field $E_{\mathrm{ferro}}$ gives rise to a voltage drop over the ferroelectric capacitor that is equal, but with opposite sign, to the

voltage drop over the serial capacitor (see the section "Methods"). The $D$-$E_{\mathrm{ferro}}$ loops derived from Eq. (3) are presented in Fig. 1c for PZT and Fig. 1f for P(VDF–TrFE). The reconstructed loops for various $C_{\mathrm{ferro}}/C_{\mathrm{ser}}$ ratios are similar to that of the ferroelectric-only capacitor. We note that Eq. (3) is applicable only when the leakage current is negligible. This crucial boundary condition of low leakage current is here realized by connecting a discrete ferroelectric capacitor in series with a discrete linear capacitor. Bilayer capacitors, consisting of an insulating layer on top of a ferroelectric layer, typically show charge conduction through the insulating layer and charge trapping at the interface. The measured $D$-$V_{\mathrm{app}}$ loops are then broadened instead of tilted. Typical broadened hysteresis loops have previously been reported by intentionally introducing interfacial layers[19] or by fatigue-induced delamination[20] that leads to resistive interfacial layers with threshold conduction[21].

The apparent remanent polarization, i.e. the displacement at zero applied bias, is extracted from the hysteresis loops of Fig. 1b, e and presented by the open circles as a function of the ratio $C_{ferro}/C_{ser}$ in Fig. 2. The value for the apparent remanent polarization can also be graphically determined from the hysteresis $D$-$E_{ferro}$ loop of the ferroelectric-only capacitor[22] (solid lines in Fig. 2), as described in the Supplementary Note 3. An excellent agreement is obtained.

**Dynamics of depolarization**. The suppression of polarization was hitherto investigated by measuring quasi-static hysteresis loops, where the electric field changes gradually. In the case, however, where a high applied electric field is abruptly switched off, the final polarization state might be different. The polarization in this case is initially still saturated leading to a huge depolarization field that is no longer fully compensated. The ferroelectric material is expected to depolarize. However, since the polarization is a highly non-linear function of the electric field, the final polarization state is not a priori known; one might even expect macroscopic polarization reversal as an overshoot effect.

Here, we investigate the depolarization dynamics. We applied a voltage pulse, large enough to fully polarize the ferroelectric capacitor. Then the applied voltage abruptly dropped to 0 V, and we recorded the transient of the electric displacement, $D(t)$. A schematic representation of the applied pulse and the measured response is given in the inset of Fig. 3a. At the end of the applied pulse there is a fast discharge of the induced polarization. Afterwards, the ferroelectric depolarization dominates the transient. As a typical example, a set of transients of a PZT capacitor in series with three different serial capacitors are presented in Fig. 3a. After a fast discharge of the induced polarization, the displacement only slightly decreases with time and stabilizes after a few seconds. When the displacement was stabilized we disconnected the serial capacitor and measured the first two $D$-$V_{ferro}$ hysteresis loops of the ferroelectric-only capacitor, presented in Fig. 3b. The displacement at the end of the transient, i.e. the retained polarization, is similar to the value at the start of the first loop, as indicated by the symbols in Fig. 3a, b. The second loop was measured as a reference and was found to coincide with the first loop, excluding experimental artifacts. The retained polarization is presented in Fig. 2a, b by solid circles as a function of the ratio $C_{ferro}/C_{ser}$. The values measured for PZT are similar to those obtained from the quasi-static measurements; for P(VDF–TrFE) the values slightly deviate. We therefore conclude that the final polarization state in the dynamic depolarization measurements is identical to the suppressed polarization state

reached quasi-statically. The transients for P(VDF–TrFE) are presented in Fig. 3c. The solid lines are depolarization transients calculated with the KAI formalism[12–15,23], adapted to a time-dependent electric field[24] and Merz law[10,25] (Supplementary Note 4). The good agreement implies that depolarization is due to domain switching without full macroscopic polarization reversal.

From the measured transients of displacement, $D(t)$, of Fig. 3c the field inside the ferroelectric material, $E_{ferro}$, is calculated using Eq. (3). The time-dependent internal electric field is presented in Fig. 3d for the two different capacitance ratios. Solid lines are iteratively calculated using the KAI formalism. A good agreement is obtained. Figure 3d shows that the two transients perfectly overlap. Furthermore, the final internal field, of about 50 MV/m, is equal to the coercive field. This will be explained below.

To measure the depolarization transient, we applied a voltage pulse, large enough to fully polarize the ferroelectric capacitor. The initial value of displacement was taken after the fast discharge of the induced polarization, as shown in the inset of Fig. 3a. The initial and final internal fields were then calculated from the corresponding displacement using Eq. (3) and presented in Fig. 4 as a function of $C_{ferro}/C_{ser}$. When the initial internal field is much lower than the coercive field, the initial and final internal fields are identical; there is no depolarization. However, when the initial internal field is much larger than the coercive field, the ferroelectric material depolarizes until the final internal field is stabilized at the coercive field, regardless of the ratio $C_{ferro}/C_{ser}$.

The changes in internal electric field can be understood as follows. In a ferroelectric-only capacitor the depolarization field is always fully compensated by countercharges in the electrodes; at zero applied bias the internal field is always zero. However, when a serial capacitor is connected, incomplete compensation results in a finite internal field, as given by Eq. (3). When this internal field is lower than the coercive field, the system is thermodynamically stable; there is no switching and no depolarization. If the initial internal field is larger than the coercive field, then domains will switch, leading to depolarization. The internal field decreases concomitantly with polarization. When the internal field becomes equal to the coercive field, a stable state is obtained. After this depolarization process, an internal electric field as high as the coercive field remains. We note that despite the existence of the depolarization field, there is no horizontal shift of the reconstructed loop (cf. Fig. 1c, f). The intercept with the field axis is anchored, as expected for a discrete ferroelectric capacitor in series with a linear capacitor; the circuit is a voltage divider and when the displacement is zero, the voltage over the serial capacitor is also zero.

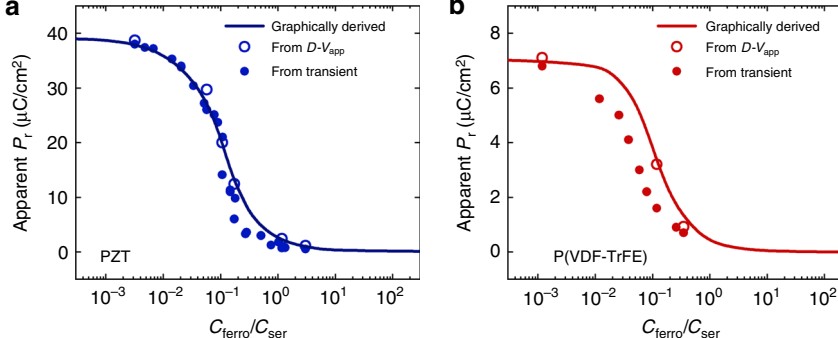

**Fig. 2** Suppression of polarization. Apparent remanent polarization, i.e. the displacement at zero applied voltage, as a function of the ratio of capacitances, $C_{ferro}/C_{ser}$, for **a** Pb(Zr,Ti)O$_3$ (PZT) and **b** poly(vinylidenefluoride–trifluoroethylene) [P(VDF–TrFE)] in serial circuits. The open circles are extracted from Fig. 1b, e. Solid circles are obtained from transient measurements described in the next section. Solid lines present graphically extracted values as discussed in Supplementary Note 3

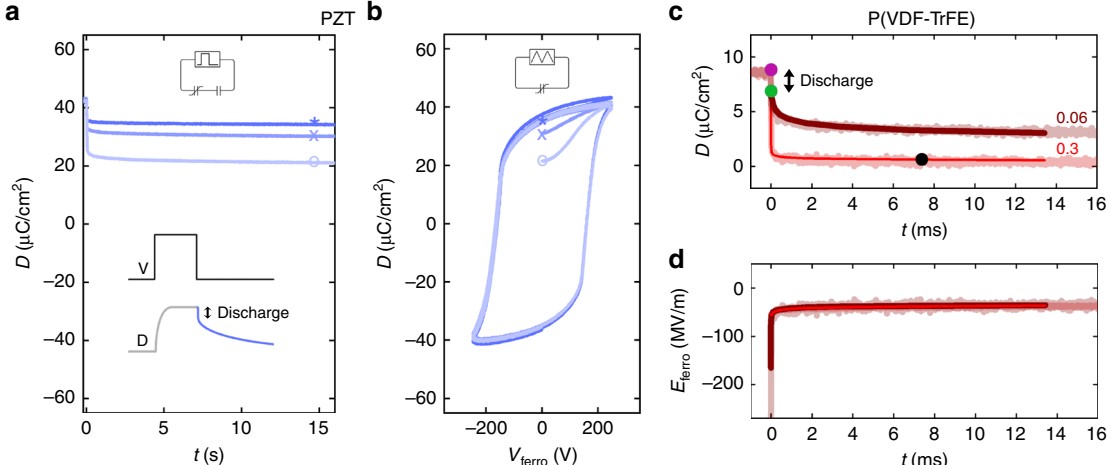

**Fig. 3 Depolarization dynamics. a** Transients of the electric displacement of a Pb(Zr,Ti)O$_3$ (PZT) ferroelectric capacitor ($C_{ferro} = 14$ nF) with various serial capacitors. The circuit and transient measurement are schematically described in the inset. The transients presented are for $C_{ser}$ as 6 µF, 330 and 110 nF, corresponding to $C_{ferro}/C_{ser}$ of 0.006, 0.04, and 0.1, respectively. The symbols (*, ×, ○) mark the value of electric displacement, i.e. the retained polarization, at the end of the transient for each capacitance ratio. **b** Reading the retained polarization by measuring the ferroelectric-only capacitor after depolarization. The measurement circuit is shown in the inset. The symbols (*, ×, ○) mark the values of the electric displacement at the end of the transients shown in **a**, demonstrating that the value of $D$ at the start of the first loop is similar to the retained polarization. **c** Displacement and **d** internal field as a function of time for poly(vinylidenefluoride–trifluoroethylene) [P(VDF–TrFE)]. The dynamics of the displacement, after a fast discharge of the induced polarization shown by the double arrow, are given by the solid lines, as obtained by numerical calculation (Supplementary Note 4). The set, discharged and final states are highlighted by the purple, green, and black circles, respectively. $C_{ferro}$ of P(VDF–TrFE) here is 260 pF and the curves correspond to serial capacitances of 4.4 nF and 890 pF, with the respective capacitance ratios indicated

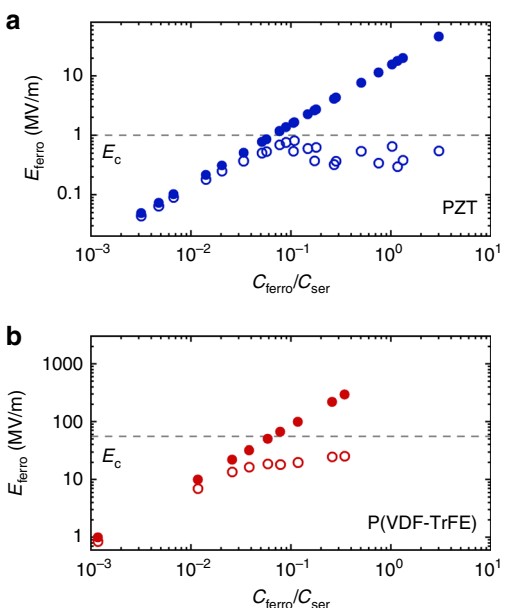

**Fig. 4 Internal electric field.** Internal electric field as a function of the ratio of capacitances for **a** Pb(Zr,Ti)O$_3$ (PZT) and **b** poly (vinylidenefluoride–trifluoroethylene) [P(VDF–TrFE)]. The solid circles represent the initial electric field, directly after the applied voltage drops to zero and the induced polarization is discharged. The open circles represent the retained values of the field after depolarization. The dashed lines represent the corresponding coercive field. When the initial internal field is much lower than the coercive field, there is no depolarization and the initial and final internal fields are identical. When the initial internal field is much larger than the coercive field, the ferroelectric material depolarizes until the final internal field is stabilized at the coercive field, regardless of the ratio $C_{ferro}/C_{ser}$

**Depolarization diagram**. To summarize the dynamic measurements, we replot the retained displacement as a function of $C_{ferro}/C_{ser}$ in the depolarization diagram of Fig. 5a. Recapitulating, for a given ratio of capacitances we apply a voltage pulse to set an initial polarization state. Then we remove the bias by grounding the electrode. From the depolarization transient, cf. Fig. 3a, c, we extract the final polarization state. In order to compare different ferroelectric materials, the displacement, cf. Fig. 2, is normalized by the remanent polarization of the ferroelectric-only capacitor, and presented in Fig. 5a as a function of $C_{ferro}/C_{ser}$. The red dots represent the normalized, retained displacement of P(VDF–TrFE) and the blue dots those of PZT. The dots overlap. The depolarization measurements of Fig. 4, have shown that in the final depolarized ferroelectric material an internal electric field as high as the coercive field remains, meaning that $E_{ferro}$ is then equal to $E_c$. Hence, in the case of depolarization, the value of the normalized retained displacement is derived from Eq. (3) as

$$D/P_r = \frac{\varepsilon_0 \varepsilon_{ferro} E_c / P_r}{C_{ferro}/C_{ser}} \qquad (4)$$

The fact that the normalized retained displacement for P(VDF–TrFE) and PZT overlaps, means that $D/P_r$, is a unique linear function of $(C_{ferro}/C_{ser})^{-1}$, with the same value of the proportionality constant, $P_r/\varepsilon_0 \varepsilon_{ferro} E_c$, here of about 15.

We note that the plotted measurements in Fig. 5a, represent the boundary of thermodynamically stable initial polarization states. The dark gray area at the top of the diagram represents fully saturated initial states reached at very high applied bias. The displacement is equal to the saturated ferroelectric polarization plus the linear displacement, or induced polarization. Upon removing the applied bias the induced polarization is discharged and the displacement reduces to the remanent polarization of the ferroelectric-only capacitor, as indicated by the horizontal gray line. At small values of $C_{ferro}/C_{ser}$ this polarization is thermodynamically stable. However, at large values of $C_{ferro}/C_{ser}$, in the

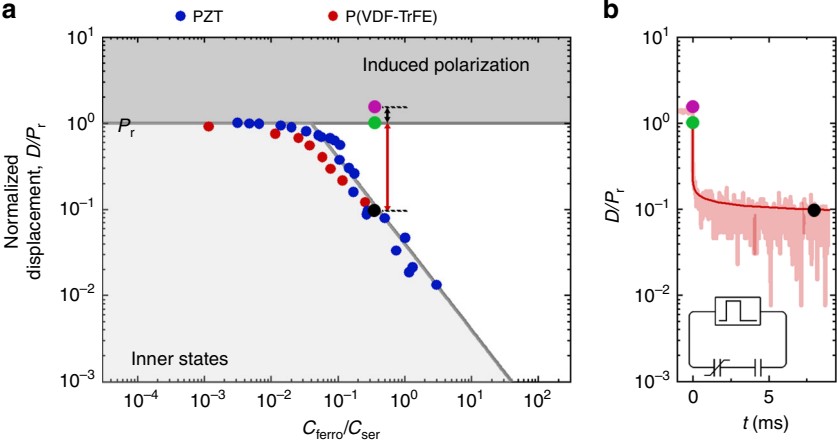

**Fig. 5** Depolarization diagram of ferroelectric materials. **a** The normalized displacement, i.e. the displacement divided by the remanent polarization of the ferroelectric-only capacitor, presented as a function of the ratio of capacitances, $C_{ferro}/C_{ser}$. The red and blue dots are the retained, normalized polarization of poly(vinylidenefluoride–trifluoroethylene) [P(VDF–TrFE)] and Pb(Zr,Ti)O$_3$ (PZT), respectively, obtained from the final state of depolarization transients. The values are replotted from Fig. 2. These dots represent the boundary, Eq. (4), between thermodynamically stable and depolarizing initial states. The dark gray area at the top of the diagram represents fully saturated initial states reached at very high applied bias. Upon grounding the electrodes, the induced polarization is discharged. The displacement gets equal to that of the ferroelectric-only capacitor, indicated by the horizontal gray line. At small values of $C_{ferro}/C_{ser}$ the polarization is thermodynamically stable. However, at large values of $C_{ferro}/C_{ser}$, in the white regions of the diagram, the polarization cannot be maintained, and the state depolarizes. The light gray area in the bottom of the depolarization diagram represents initial polarization states formed by incomplete or partial switching of the polarization. These inner states are thermodynamically stable and for a given displacement the internal field increases with $C_{ferro}/C_{ser}$ and reaches the coercive field at the boundary. **b** As an illustration we replot a normalized transient for P(VDF–TrFE) from Fig. 3c. The light-colored area corresponds to experimental data, while the solid lines describe the dynamics of depolarization as obtained by numerical calculation (Supplementary Note 4). The purple circle denotes the set initial state and the black arrow indicates the measured discharge of the induced polarization. Subsequently, the discharged state corresponding to the green circle depolarizes, as indicated by the red arrow. The final state is indicated by the black circle

white regions of the diagram of Fig. 5a, the polarization cannot be maintained, and the state depolarizes. This depolarization process is corroborated by replotting a transient for P(VDF–TrFE) from Fig. 3c next to the depolarization diagram in Fig. 5b. The set, discharged and final states are highlighted by the purple, green and black circles, respectively. Upon shorting the electrodes first the induced polarization is discharged. The remaining displacement is then equal to the remanent polarization of the ferroelectric-only capacitor. However, the ferroelectric material depolarizes, as the remanent polarization is suppressed by the serial capacitor. After depolarization the remanent polarization in the final state is equal to the boundary value of the displacement at that ratio of capacitances. We note that the final internal field is equal to the coercive field and that the depolarization transient features a negative differential capacitance, the absolute value of which is equal to the serial capacitance during the whole depolarization process (Supplementary Note 5).

The light-gray area at the bottom of the depolarization diagram shown in Fig. 5a represents states where the initial set displacement is smaller than the maximum remanent polarization of the ferroelectric only capacitor. Basically, this area represents the inner polarization states formed by incomplete or partial switching of the polarization. The transient measurements only show discharge of the induced polarization. There is no depolarization as the inner polarization states are stable over time, up to the Curie temperature[26]. The stability originates from the coexistence of effectively independent domains, with slightly different values of polarization and coercive field[26]. In the final state, the internal field is zero when the ferroelectric capacitance is much smaller than the serial capacitance. For a given electric displacement the internal field increases with $C_{ferro}/C_{ser}$ and reaches the coercive field at the boundary.

**Universality of $P_r/\varepsilon_0\varepsilon_{ferro}E_c$.** Figure 5a shows that the normalized displacement for P(VDF–TrFE) and PZT capacitors, retained after a pulse excitation, is a unique linear function of $(C_{ferro}/C_{ser})^{-1}$. The proportionality constant is $P_r/\varepsilon_0\varepsilon_{ferro}E_c$ and a value of about 15 was extracted. We calculated this constant for other multidomain ferroelectric materials, using reported values; particular issues such as size dependence in nanoscale materials are beyond the scope of this work. The parameters are presented in Table 1 and plotted in a parallel coordinate graph in Fig. 6. Among different ferroelectric materials the values of coercive field and dielectric constant vary by orders of magnitude. Remarkably, for the wide variety of ferroelectric materials considered here, a comparable value for $P_r/\varepsilon_0\varepsilon_{ferro}E_c$ is obtained, of about 15. The standard deviation is only 4.9, which suggests that this constant is universal.

A prerequisite for the universality of this constant, is its temperature independence, even though all comprising parameters, i.e. $P_r$, $\varepsilon_{ferro}$ and $E_c$, are strongly temperature dependent. To this end, we have measured for P(VDF–TrFE) the polarization, dielectric constant and coercive field of ferroelectric-only capacitors at temperatures between 213 and 333 K (Supplementary Note 6). In Fig. 7a, we show experimentally that for the ideal model system P(VDF–TrFE) the ratio $P_r/\varepsilon_0\varepsilon_{ferro}E_c$ is indeed temperature independent.

## Discussion

In a defect-free single crystal, polarization reversal originates from a coherent, collective rotation of dipoles, generally referred to as intrinsic switching. However, commonly used ferroelectric materials are not homogeneous, defect-free single crystals but inhomogeneous polycrystalline thin films or ceramics. The presence of defects lowers the barrier for polarization reversal. The

**Table 1 $P_r/\varepsilon_0\varepsilon_{ferro}E_c$ for multidomain ferroelectric materials**

| Ferroelectric material | Code | $P_r$ ($\mu C/cm^2$) | $E_c$ (MV/m) | $\varepsilon_{ferro}$ | $P_r/\varepsilon_0\varepsilon_{ferro}E_c$ | Reference |
|---|---|---|---|---|---|---|
| P(VDF–TrFE) 65/35 | P(VDF–TrFE) | 7 | 50 | 10 | **16** | This work |
| β PVDF | β PVDF | 7 | 60 | 10 | **13** | Own data |
| δ PVDF | δ PVDF | 7 | 115 | 8 | **9** | 39 |
| PZT 507 soft | PZT507 | 40 | 1.0 | 3500 | **13** | This work |
| PZT 4 hard | PZT4 | 25 | 1.28 | 1260 | **18** | Own data |
| PbTiO$_3$ | PT | 19 | 7 | 300 | **10** | 40 |
| BaTiO$_3$ | BTO SPS | 12.4 | 0.23 | 3686 | **20** | 41 |
| BaTiO$_3$ | BTO | 20 | 0.37 | 4250 | **15** | 42 |
| SrBi$_2$Ta$_2$O$_9$ | SBTO | 10.5 | 3.0 | 400 | **10** | 43 |
| Nylon-11 | Nylon11 | 5.1 | 61 | 4.8 | **20** | 44 |
| Nylon-7 | Nylon7 | 8.5 | 80 | 5 | **24** | 44 |
| trialkylbenzene-1,3,5-tricarboxamide | BTA-C10 | 4 | 30 | 8 | **19** | 45 |
| trialkylbenzene-1,3,5-tricarboxamide | BTA-C18 | 2.5 | 20 | 8 | **18** | 45 |
| Bi$_{0.5}$Na$_{0.5}$TiO$_3$ | BNT1 | 38 | 7.3 | 350 | **17** | 46 |
| Bi$_{0.5}$Na$_{0.5}$TiO$_3$ | BNT2 | 35 | 7.0 | 400 | **14** | 47 |
| Bi$_{0.5}$Na$_{0.5}$TiO$_3$–Bi$_{0.5}$K$_{0.5}$TiO$_3$ | BNKT50 | 25 | 3.0 | 836 | **11** | 47 |
| Bi$_{0.5}$Na$_{0.5}$TiO$_3$–Bi$_{0.5}$K$_{0.5}$TiO$_3$ | BNKT80 | 35 | 2.5 | 1640 | **10** | 47 |
| (Na$_{0.535}$K$_{0.485}$)$_{1-x}$Li$_x$(Nb$_{0.8}$Ta$_{0.2}$)O$_3$ | NKL$_x$NT | 22.7 | 1.46 | 1091 | **18** | 48 |
| Sr$_{0.15}$(Na$_{0.5}$Bi$_{0.5}$)$_{0.85}$TiO$_3$ | SNBTO | 27 | 2 | 1000 | **15** | 49 |

The (bold) values of $P_r/\varepsilon_0\varepsilon_{ferro}E_c$ are calculated using reported values for the remanent polarization, $P_r$, coercive field, $E_c$, and dielectric constant, $\varepsilon_{ferro}$. For P(VDF–TrFE) the values are obtained at ambient temperature. In Fig. 7a we show that the value of $P_r/\varepsilon_0\varepsilon_{ferro}E_c$ remains constant between 213 and 333 K. For BTO SPS values were averaged over six spark-plasma-sintered (SPS) samples, with an average grain size ranging between 0.8 and 18.5 μm. For BTO values were averaged over four samples, with an average grain size ranging between 1.23 and 75.6 μm. For NKL$_x$NT values were averaged over seven samples of varying composition ($0.02 \leq x \leq 0.08$)

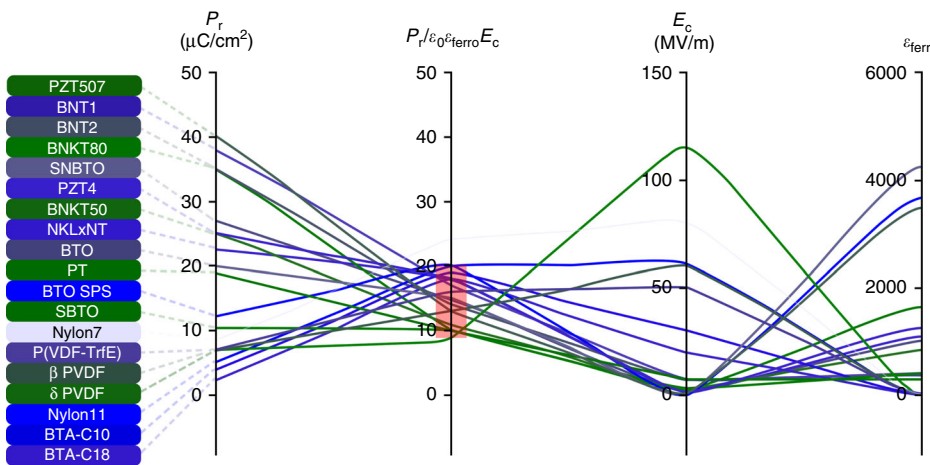

**Fig. 6** Parallel coordinate plot of $P_r/\varepsilon_0\varepsilon_{ferro}E_c$. The remanent polarization, $P_r$, coercive field, $E_c$, dielectric constant, $\varepsilon_{ferro}$, and the ratio $P_r/\varepsilon_0\varepsilon_{ferro}E_c$, for various ferroelectric materials, as summarized in Table 1, are presented in parallel coordinates. Lines are guides to the eye and are colored in a gradient from green to blue for clarity. Despite the fact that the values of coercive field, remanent polarization and dielectric constant vary by several orders of magnitude, for all ferroelectric materials considered here a comparable value for $P_r/\varepsilon_0\varepsilon_{ferro}E_c$ is obtained

collective rotation of dipoles is replaced by nucleation and anisotropic growth of individual domains, termed as extrinsic switching. In the discussion below, we first show that despite apparent similarities in the relation between coercive field, polarization and dielectric constant, the physical mechanism of intrinsic switching is completely different from the nucleation and growth studied here for extrinsic switching. We therefore focus on extrinsic switching by domain-wall motion. We then experimentally derive a linear relation between activation field and coercive field. The relation is supported by the classical theory of Miller and Weinreich[27], and by molecular dynamic calculations[11,28]. Using the experimentally extracted universal value of $P_r/\varepsilon_0\varepsilon_{ferro}E_c$ of about 15, we connect the activation field with the depolarization field. We show that the depolarization

field is the onset of domain-wall flow and similar to the activation field, $E_{dep} \sim E_{act}$.

**Intrinsic switching.** Intrinsic switching occurs at the intrinsic, or thermodynamic, coercive field $E_{int,c}$ where the Landau double potential well is destroyed. In the seminal work of Tagantsev et al., a relation between the intrinsic coercive field, the polarization $P$ and the dielectric constant, has been theoretically derived as (ref. [15], Eq. 2.3.15, p. 85)

$$\frac{P}{\varepsilon_0\varepsilon_r E_{int,c}} = 3\sqrt{3} \approx 5 \quad (5)$$

Details of the derivation are given in the Supplementary Note 7. The relation is reminiscent of our experimentally derived

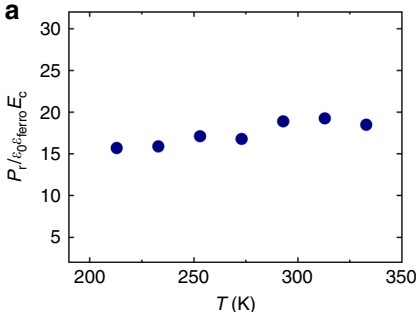
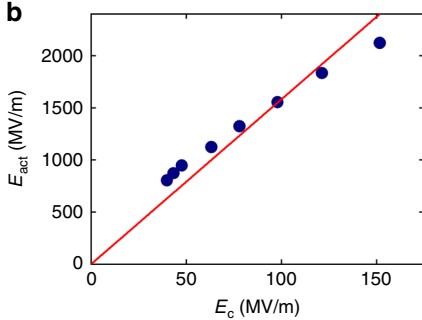

**Fig. 7** Extracted ferroelectric parameters for the ideal model system P(VDF–TrFE). **a** Extracted value of $P_r/\varepsilon_0\varepsilon_{ferro}E_c$ as a function of temperature for poly (vinylidenefluoride-trifluoroethylene) [P(VDF–TrFE)]. **b** Activation field as a function of coercive field extracted from switching measurements on P(VDF–TrFE). The red line is a linear fit through the origin, resulting in a slope of ~15

relationship, viz. $P_r/\varepsilon_0\varepsilon_{ferro}E_c \approx 15$. However, the physical mechanisms are completely different, and cannot a priori be related.

Intrinsic switching is a mean-field process, which dictates collective behavior of the dipoles within the ferroelectric material. This happens when all dipoles are in a homogeneous environment; a necessary condition is that there are no pinning defects, which is practically unrealistic. Nevertheless, at very high electric field, the electrostatic energy dominates, thus influences from defects may become trivial: the pinning force by the defects is overwhelmed by the applied high field; the total electric field that the dipoles feel, either in defect-free regions or in vicinity of pinning defects, are about the same, and the system is effectively a homogeneous system. In that case, even a practical ferroelectric material with a considerable number of pinning defects may favor undergoing intrinsic switching.

Experimentally approaching the intrinsic switching has being a great challenge. To unambiguously correlate the depolarization and activation field with the intrinsic coercive field, the comprising physical constants of the same device should be measured. Hence, P(VDF–TrFE) is an ideal model system as the dielectric constant hardly depends on microstructure and film thickness, and the complete data set on depolarization-, activation- and intrinsic coercive field is available. We show (cf. Supplementary Note 7), that the activation field, depolarization field and intrinsic coercive field are of the same order of magnitude, $E_{dep} \sim E_{act} \sim E_{int,c}$.

However, we note that this equality can be artificial as the underlying physical mechanisms for intrinsic and extrinsic switching are completely different. Current experiments, in our work as well as in literature, are insufficient to draw a solid conclusion about this correlation. For this reason, we restrict our discussion to the activation field and depolarization field.

**Relation between activation field, $E_{act}$, and coercive field, $E_c$.** We note that $E_{act}$, as it appears in Merz's law, viz. Eq. (2), is not a real electric field but a phenomenological parameter with units of electric field, which is related to the energy required to move the domain walls. Here we argue that the energy barrier to overcome for domain-wall motion corresponds to an actual electric field, i.e. the depolarization field.

Merz's law is observed in many ferroelectric systems ranging from single crystals through bulk ceramics[29], and thin films[9,30], to organic-ferroelectric composites[31]. We note that Tybel et al.[8] were the first to point out that Merz's law is a special case of domain-wall motion in generic creep systems, describing propagation of elastic objects driven by an external force in

the presence of a pinning potential, such as domains in ferroelectric[9] and magnetic materials[32] and vortices in type-II superconductors[33].

A few papers have addressed the relation between activation field and coercive field. A phenomenological expression has been suggested[34]:

$$E_c \approx \frac{E_{act}}{\ln(\gamma E_c) - \ln(16f)} \tag{6}$$

where $\gamma$ may be regarded as the displacement velocity of domains per volt and $\gamma E_c$ has a large value and can be approximated as a constant. The coercive field then depends linearly on activation field.

Here we use P(VDF–TrFE) to experimentally derive the relation between activation field, coercive field, and polarization. We emphasize that P(VDF–TrFE) is a unique model system. Contrary to most inorganic ferroelectric materials, the switching time for P(VDF–TrFE) can be measured over a wide range of electric field and temperature. The details of the measurements and data extraction are reproduced from ref. [24] in the Supplementary Note 6. The extracted activation field as a function of the quasi-static coercive field is presented in Fig. 7b.

Figure 7b shows that the activation field linearly increases with the value of the coercive field: $E_{act}/E_c \approx 15$. A similar relation can be derived from the experimental data reported in ref. [23]. The relation between and activation and coercive field has also theoretically been addressed[11,27,28]. The classical theory supporting Merz's law was developed by Miller and Weinreich[27], derived for 180° domain-wall motion in BaTiO₃. The activation field is not an electric field, but a parameter related to the domain-wall energy. The rate determining process is the nucleation of steps along the domain wall. The nucleation energy, $\Delta U^*$, is determined by the competition between the cost of creating additional domain walls of the nucleus and the gain of the alignment with the applied electric field:

$$\Delta U^* = \frac{c\sigma_{dw}^2}{P_{sat}E_c} \tag{7}$$

where $c$ is the width of the domain wall, $\sigma_{dw}$ is the domain-wall energy, and $E_c$ is the coercive field. Subsequently, Miller and Weinreich arrived at an expression for the activation field that has been reformulated as[23]

$$E_{act} = \frac{c\sigma_{dw}^2}{P_{sat}k_B T} \tag{8}$$

By eliminating $c\sigma_{dw}^2$ we arrive at a relation between $E_{act}$ and $E_c$ as

$$\frac{E_{act}}{E_c} = \Delta U^* / k_B T \qquad (9)$$

A range of values for $\Delta U^*$ has been reported[23], e.g. $10 k_B T$ for PZT[35], $15 k_B T$ for PVDF Langmuir–Blodgett films[36], $29 k_B T$ for thin films of P(VDF–TrFE), and $40 k_B T$ for BaTiO$_3$[37], leading to a value for the ratio of $E_{act}$ over $E_c$ in the range of 10–40.

We note that Miller and Weinreich suggested that the critical nucleus is an atomically thin triangular plate with a large aspect ratio, which then expands laterally on the same atomic plane. However, it is well-established that the Miller–Weinreich theory overestimates the activation field by an order of magnitude[11]. Molecular dynamics simulations for 180° domain walls in defect-free PbTiO$_3$ did reveal not a triangular but a square critical nucleus with diffusive and beveled interfaces that substantially reduces the nucleation barrier and hence leads to much lower activation fields for domain-wall motion. Nevertheless, here we are not calculating the absolute value of the activation field, but we only consider the ratio of $E_{act}$ over $E_c$ using experimentally extracted values of the nucleation energy. The proportionality constant is comparable to the one experimentally derived for P(VDF–TrFE), and agrees with molecular dynamics simulations[11,28].

**Relation between activation field, $E_{act}$ and depolarization field, $E_{dep}$.** The values for $E_{act}/E_c$ and $P_r/\varepsilon_0\varepsilon_{ferro}E_c$ are comparable. Therefore, we suggest that $E_{act}$ is comparable to $P_r/\varepsilon_0\varepsilon_{ferro}$. As $P_r/\varepsilon_0\varepsilon_{ferro}$ is by definition equal to the depolarization field, $E_{dep}$, this then suggests that $E_{act} \sim E_{dep}$.

This relation can tentatively be explained as follows. In multidomain ferroelectric materials, pinning sites induce a disordered local field. At zero temperature the domain walls are pinned. The domain walls undergo a pinning/depinning transition when sufficient energy is provided by the electrostatic field. However, at finite temperature, even strongly pinned domain walls can propagate due to the thermal energy. Hence, the domain-wall motion at electric fields even above $E_c$ is described by creep, i.e. thermally activated motion. With increasing electric field, the relative contribution of the electrostatic energy increases concomitantly. At the activation field the electrostatic energy dominates. Then the domain-wall motion is in the flow regime; the velocity is independent of the temperature and is linearly dependent on the electric field[28]. Hence, the activation field corresponds to the threshold for the transition between domain-wall creep and flow. Here we consider pinning sites as dipoles with a fixed polarity, i.e. their dipole moment remains frozen for all electric fields. Dipoles in the vicinity tend to align parallel to the polarity of the pinning site. Domain-wall flow then implies switching of these pinned polarized regions. We note that the average electric field within these polarized regions is the depolarization field, which has to be overcome in order to move the domain walls. Hence the depolarization field is similar to the activation field, $E_{dep} \sim E_{act}$.

In conclusion, we have investigated the dynamics of depolarization in ferroelectric materials. To tune the compensation of the depolarization field we used a ferroelectric capacitor in series with linear capacitors. The stability of any set polarization state is summarized in a depolarization diagram that shows depolarizing and thermodynamically stable regimes. The boundary separating the two regimes is obtained when the internal electric field is equal to the coercive field, and yields a unique relation among the coercive field, dielectric constant and remanent polarization, $P_r/\varepsilon_0\varepsilon_{ferro}E_c$. This experimentally derived constant is similar for PZT and P(VDF–TrFE) and equal to about 15. Among a large number of different ferroelectric materials the values of coercive field, dielectric constant, and polarization vary by orders of magnitude. Remarkably however, the values for $P_r/\varepsilon_0\varepsilon_{ferro}E_c$ are comparable, which indicates a universal character of the extracted constant. Using the experimentally extracted universal value of $P_r/\varepsilon_0\varepsilon_{ferro}E_c$, we draw a link between the depolarization field and the activation field for domain-wall depinning in disordered ferroelectric systems. We argue that the depolarization field is the onset of domain-wall flow and, thus, similar to the activation field, $E_{dep} \sim E_{act}$.

## Methods

**Experimental**. Ferroelectric capacitors were based on the random copolymer P (VDF–TrFE) (composition 65%/35%) and on soft PZT. Capacitors were prepared as described previously[24]. The P(VDF–TrFE) films with a thickness of about 500 nm were spin-coated from a 5 wt% solution in methylethylketone on thermally oxidized silicon monitor wafers, on which 50 nm-thick Au bottom electrodes on a 2 nm Ti adhesion layer were photo-lithographically defined. To enhance the crystallinity and hence the ferroelectric properties, the samples were subsequently annealed in vacuum at 140 °C for 2 h. To finalize the P(VDF–TrFE) capacitors, a Au top electrode was evaporated through a shadow mask. Soft PZT ceramics (PZT507) were obtained from Morgan Advanced Ceramics. The thickness was 200 μm. Capacitances were measured with Solartron SI 1260 impedance/gain-phase analyzer. Serial capacitors (WIMA GmbH, Germany) were selected for low dielectric losses. The ferroelectric hysteresis loops were measured with a Radiant Precision Multiferroic Test System (Radiant Technologies, Inc.). The frequency was set at 1 Hz, to enable quasi-static measurements. The polarization transients were measured with a Sawyer–Tower circuit consisting of a Tektronix AFG3102 function generator, a LeCroy waverunner LT372 oscilloscope, and a Krohn-Hite 7602 M wide-band amplifier.

**Tuning the internal electric field by a serial capacitor**. In a conventional Sawyer–Tower circuit, a voltage, $V_{app}$, is applied to a ferroelectric capacitor and subsequently the electric displacement, $D$, of the ferroelectric material is measured over a reference capacitor. The internal field in the ferroelectric material, $E_{ferro}$, is derived as the applied voltage divided by the thickness, $d$, of the ferroelectric material.

To study the suppression of polarization we use a linear capacitor, $C_{ser}$, in series with the ferroelectric capacitor, $C_{ferro}$. We analyzed the electrical circuit using classical electrostatics. The applied voltage is shared by the linear capacitor and the ferroelectric capacitor, and reads:

$$V_{app} = V_{ferro} + V_{ser} \qquad (10)$$

where $V_{ferro}$ and $V_{ser}$ are the voltage drops over the ferroelectric- and serial capacitor, respectively. The values can be given as

$$V_{ferro} = \frac{Q_{ferro} - PA}{C_{ferro}} \text{ and } V_{ferro} = V_{app} - \frac{Q_{ser}}{C_{ser}} \qquad (11)$$

where $P$ and $A$ are the ferroelectric polarization and the area of the ferroelectric capacitor, respectively. Within a single metallic electrode there cannot exist an electric field. According to the Gaussian theorem, the total charge at the surface of the electrode, $Q_{ferro}$, is given by the surface integral $\varepsilon_0 \oiint \mathbf{E} \cdot d\mathbf{S}$, where $\varepsilon_0$ is the permittivity of free space, $\mathbf{E}$ is the electric field, and $d\mathbf{S}$ the infinitesimal vector normal outside from the surface. As there is no current flowing, $E$ is zero at all points on the surface, hence the net charge inside the boundary is zero. Therefore, electro-neutrality holds, i.e. $Q_{ferro} = Q_{ser} = V_{ferro}C_{ferro} + PA$. We then get:

$$V_{ferro} = \frac{C_{ser}}{C_{ser} + C_{ferro}} V_{app} - \frac{PA}{C_{ser} + C_{ferro}} \qquad (12)$$

We note that this equation is identical to Eq. (2) in ref. [3]. Experimentally, the electric displacement, $D$, is directly extracted from the Sawyer–Tower measurements, and is regarded as fundamental variable[38]. To obtain the internal field within the ferroelectric material, we eliminate the ferroelectric polarization, $P$, by noting that:

$$D = \frac{\varepsilon_0 \varepsilon_{ferro} V_{ferro}}{d} + P \qquad (13)$$

As the area of the ferroelectric capacitor, $A$, is given by $C_{ferro}d/\varepsilon_0\varepsilon_{ferro}$, the electric field inside the ferroelectric material is derived as

$$E_{ferro} = \frac{V_{app}}{d} - \frac{D}{\varepsilon_0\varepsilon_{ferro}} \cdot \frac{C_{ferro}}{C_{ser}} \qquad (14)$$

This relation is reproduced as Eq. (3) in the main text of the manuscript. Using the expression for the depolarization field, cf. Eq. (1), and taking Eq. (14), we get:

$$E_{ferro}(t) = E_{dep} \frac{1}{1 + (C_{ferro}/C_{ser})^{-1}} = -\frac{P(t)}{\varepsilon_0\varepsilon_{ferro}} \cdot \frac{1}{1 + (C_{ferro}/C_{ser})^{-1}} \qquad (15)$$

As compared to a single ferroelectric capacitor, the internal field relates to the depolarization field, $E_{dep}$, tuned by the ratio of the capacitances of the ferroelectric-

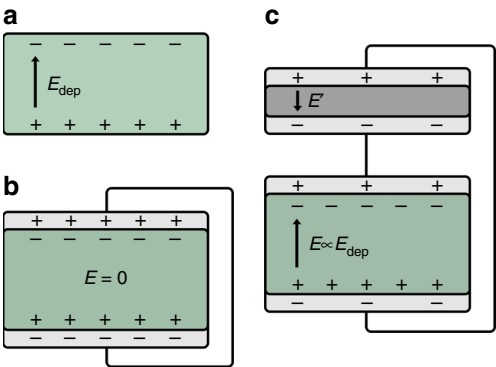

**Fig. 8** Schematics of compensation of depolarization field. **a** Depolarization field in a free-standing ferroelectric material. **b** Ferroelectric-only capacitor with metal electrodes. **c** Ferroelectric capacitor with an additional serial capacitor

and the linear serial capacitor, $C_{ferro}/C_{ser}$. This net field by a serial capacitor is also schematically indicated in the sketch of Fig. 8.

For a ferroelectric-only capacitor the charges in the electrodes fully compensate the polarization, such that at zero bias there is no net field in the ferroelectric material. In case of an additional serial capacitor, where due to charge neutrality the amount of free charges on both capacitors is equal, the depolarization field is not fully compensated. This leads to a net field inside the ferroelectric capacitor that is opposite to the electric field in the serial capacitor. At zero bias therefore the voltage drop over both capacitors is equal, but of opposite sign, such that the total voltage is zero. The smaller the capacitance of the serial capacitor the higher the net field in the ferroelectric material will become, as demonstrated by Eqs. (14) and (15).

**Transient displacement measurements**. We first fully depolarized the ferro-electric capacitor by applying an alternating voltage with decreasing amplitude[26]. Subsequently the linear capacitor was discharged by shorting the electrodes, to ensure electro-neutrality between the ferroelectric- and linear capacitor. Then we applied a square pulse over the serially connected capacitors. The amplitude was large enough to fully polarize the ferroelectric capacitor. Contrary to quasi-static Sawyer–Tower measurements, the applied voltage was abruptly set to zero volt and the resulting transient of the electric displacement, $D(t)$, was recorded.

## Data availability
The data that support the findings of this study are available from the corresponding author upon reasonable request.

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

## Acknowledgements

The authors acknowledge financial support from the Max-Planck Institute for Polymer Research (Mainz, Germany). We gratefully acknowledge Dr. Kamal Asadi and Prof. Paul W. Blom for fruitful discussions, and Mr. Hanspeter Raich and Mr. Benjamin Zwietasch for technical support, all from the Max-Planck Institute for Polymer Research. We thank Mr. Cheng Guo from Peking University for research assistance.

## Author contributions

D.M.d.L. and D.Z. conceived the idea and designed the experiments. T.L. and P.G. fabricated the devices. D.Z. performed the measurements and analysis. I.K. and D.D. collaborated with the analysis. D.Z., T.L., G.H.G., P.G., D.M.d.L. and I.K. co-wrote and commented on the manuscript. I.K. supervised the project.

## Additional information

**Competing interests:** The authors declare no competing interests.

**Journal Peer Review Information:** *Nature Communications* thanks the anonymous reviewers for their contribution to the peer review of this work. Peer reviewer reports are available.

