## [Peer Review File · Nature Communications]

Reviewers' comments:

Reviewer #1 (Remarks to the Author):

The manuscript presents interesting experimental data that appear to be of high quality. My only reserves concern the conceptual organisation of the material, as the manuscript is not always clear about what models are used to interpret the results.

First of all, the depolarisation field is not a bulk property of the ferroelectric, as it depends on the electrical boundary conditions. The authors attempt a boundary-independent definition of E_{dep} in Eq.(1), but this expression doesn't make too much sense to me, neither in an experimental or theoretical context. How would the authors calculate this quantity from first principles, for example? What would be its physical interpretation?

A much more meaningful (and theoretically well defined) quantity to look at is the "intrinsic coercive field", which the authors discuss as well. This quantity can be calculated, either in a Landau or first-principles context, and related to measurable quantities (P_r and ϵ_{ps_ferro}) via Eq.(10). I believe that the manuscript would much improve in clarity if the authors stated Eq.(10) upfront instead of their Eq.(1), possibly by giving some hints about its derivation and the assumptions that were made. Then say that they estimate $E_{c,int}$ via a measurement of ϵ_{ps_ferro} and P_r . There is no need whatsoever to speak about the confusing and ill-defined E_{dep} , since $E_{c,int}$ is directly related to it.

The second quantity the authors look at is the quasistatic coercive field. This is a known quantity and techniques for measuring it well established. The third quantity is the activation field, E_{act} . I guess that this is the quantity that is hardest to measure (via depolarization dynamics), and the real conceptual/technical value of this work. The authors should state upfront that the largest part of the experimental effort, and the most remarkable novelty of this work, is the accurate measurement of E_{act} . Of course, the authors do not measure directly E_{act} , but only indirectly via some models and assumptions -- these should be clearly stated (i.e. what do the authors measure and what formula do they use to estimate E_{act} and why?)

Then, after the authors have these three quantities measured and the techniques for doing that well detailed, only then they should proceed at discussing their relationship, and at drawing some reasonable conclusions about the physical mechanism that is involved in switching. Right now all this material (discussions, speculations, etc.) is mixed up and I had to read the manuscript several times to figure out what is being done and how.

After the major restructuring that I suggest above, I guess this could be a nice paper. I have a question on the physics, actually. The intrinsic coercive field assumes a monodomain switching passing through the saddle point of the Landau potential. This might be a well justified scenario in uniaxial ferroelectrics, but how about PZT? I guess that PZT might find some alternative paths, with substantially smaller barriers, by rotating the polarization vector, and therefore avoiding the very high-energy saddle point. Did the authors find a systematically smaller value of their universal constant in uniaxial ferroelectrics, by any chance?

Reviewer #2 (Remarks to the Author):

In their paper "Depolarization of multidomain ferroelectric materials" Katsouras et al. study domain wall depinning and depolarization in a dynamic regime. They especially infer a single constant to describe the boundary between a stable ferroelectric state and a region where depolarization governs. The effect of depolarization and how to engineer and understand stable ferroelectric states is important and has been a field of large research for more than 40 years. In this work the authors study these effects in an electric circuit setting including series capacitances,

whilst traditionally one has tried to investigate in proper short circuit conditions.

I do have some concerns regarding the paper that should be clarified to better convey the message to the readers:

1. The authors perform a lump circuit electrostatic analysis. In such an analysis they do not take into account the effect of the finite screening of the electrode. It has been shown theoretically that due to a finite screening length of the electrodes the internal field will never be zero inside a ferroelectric, even under short circuit conditions. How does this affect the statements made in the paper of a regime with zero depolarization?
2. In the experiments the authors use the value of the series capacitance as control parameter, whilst the ferroelectric capacitance is constant. When changing the series capacitor, the system will also exhibit a change of effective RC time constant, and a saturation would be expected for small series capacitances, that is in the same region where the authors claim a thermodynamic stable state? Have the authors tried to perform similar measurements with C series constant and effectively changing the ferroelectric capacitance systematically?
3. The authors claim that the constant $P/(\epsilon_0 \epsilon_f E_c)$ is a universal constant, however in the paper they show data where the constant varies within a factor 3-4. Moreover, they claim that the constant is also valid close to a phase transition (table 2), this is not clear from the theoretical part of the paper why it should be valid close to a phase transition? If a universal constant one would assume it to be the same for a given material, and maybe deviate close to phase transitions?

Taken together, the authors have presented a solid study, however I feel that the presented data and analysis is better suited for a more specialized journal when the points above are addressed.

Reviewer #3 (Remarks to the Author):

The work described in the paper is quite interesting. The authors use an ingenious method to apply an electric field through the depolarization mechanism and use it to derive a universal relationship between the remanent polarization, dielectric constant and coercive field of ferroelectric materials. While I think the experimental results are quite important and general, I have some questions about their interpretation by the authors. The authors argue that the activation field of Merz's law is the depolarization field and ascribe this to depinning from defects. They use the original model proposed by Miller and Weinreich to explain the origin of the activation field and also try to link their result for the coercive field being proportional to P/ϵ_0 to the intrinsic coercive field as described by Tagantsev et al. I think this is either explained in a confusing way or a mistake.

First, the intrinsic coercive field derived in the book by Tagantsev et al. is the field at which one of the double wells in the FE double well potential is destroyed. At this field, the switching process is intrinsic, that is to say the entire material will switch simultaneously on the ps timescale. This is clearly not relevant to the extrinsic, nucleation-and-growth driven domain wall motion process that proceeds on much slower timescales precisely because the different domains are located in the different sides of the double well potential energy surface.

Second, the extrinsic DW motion process is thought to be controlled by the nucleation energy which is determined by the competition between the cost of creating additional domain walls of the nucleus and the gain of the alignment with the applied E field, as described first by the Miller and Weinreich model cited by the authors. Thus, the coercive field is (roughly) determined by the ratio of the square of the domain wall energy and polarization (σ^2/P). The authors find that the coercive field is proportional to P/ϵ_0 . Perhaps the σ^2/P ratio of the Miller-Weinreich

model is proportional to the P/ϵ ratio derived by Tagantsev for the intrinsic coercive field and experimentally by the authors for the measured coercive field, but it is not clear why these two quantities (σ^2/P) and (P/ϵ) should be linearly related and whether for example they have a similar temperature dependence. The authors should demonstrate the equivalence between these two formulas for E_c . Alternatively, if these two formulas are not equivalent, this would indicate that the Miller-Weinreich model is inapplicable for the switching studied in this work. The finding that E_c does not follow the standard nucleation model would be quite unusual and would lead to the question of what mechanism governs the switching observed by the authors and what relationship this mechanism has to domain wall motion.

Third, the activation field in Merz's law is not really a field in the same sense as an applied electric field or the depolarizing field. The activation field is merely a bookkeeping device that collects all of the constants in the Miller-Weinreich formula or alternatively a parameter that sets the dependence of the domain wall velocity on the applied field. Therefore, identifying the activation field with the actual existing electric field in the material (the depolarization field) is confusing. The fact that the coercive field is about ten times smaller than the activation field means that thermal energy that is ten times smaller than the barrier height is sufficient to provide the thermal fluctuations for switching to occur. It is not clear whether or how this is related to the nature of the field applied to drive the nucleation process. Thus, the statement that $E_{\text{depol}} \sim E_{\text{activation}}$ is in not very physically meaningful, because this merely says that E_{depol} has the same magnitude as $E_{\text{activation}}$ but in no way implies that E_{depol} is the field that governs the switching in all FE materials where $E_{\text{activation}}$ is observed to be ten times larger than E_c .

Finally, the authors use the results of previous P_r , ϵ and E_c measurements to demonstrate the universality of the relationship between E_c and P/ϵ . If these data were taken from the measurements using the standard electrodes, would the authors approach for estimating E_{depol} be still valid, considering that in the presence of electrodes, E_{depol} depends on the screening length of the electrodes and on the thickness of the ferroelectric? The authors should explain this.

Because of these questions, I cannot recommend the paper for publication in its current form. However, if the authors would address these points, I think the paper is likely to be suitable for publication in Nature Communications.

Comments of Reviewer #1 and answers by the authors:

The manuscript presents interesting experimental data that appear to be of high quality. My only reserves concern the conceptual organisation of the material, as the manuscript is not always clear about what models are used to interpret the results.

We thank the reviewer for their positive comments on the quality of our work. Their critical insights allowed us to better describe the models used, restructure the discussion over the relationship between the depolarization and activation field, and clarify their physical interpretation.

To facilitate reading of our explanations, the response is structured as follows. First all reviewer's comments are copied. Subsequently, although we agree with the reviewer that the physical mechanism of intrinsic switching is different from the nucleation and growth studied here for extrinsic switching, we include a brief discussion on intrinsic switching for clarity. We then focus on extrinsic switching by domain wall motion. We experimentally derive a linear relation between activation field and coercive field. The relation is supported by the classic theory of Miller and Weinreich^[1], and by molecular dynamic calculations^[2,3]. Then, by using the experimentally extracted universal value of $P_r/\epsilon_0\epsilon_{ferro}E_c$ of about 15, we can connect the activation field with the depolarization field. We show that the depolarization field is the onset of domain-wall flow and similar to the activation field, $E_{dep} \sim E_{act}$.

In the revised manuscript we have addressed all the points raised. Thanks to the comments of the reviewer the discussion part in the revised manuscript has improved significantly. We hope that considering the changes made, the manuscript is now suitable for publication in *Nature Communications*.

First of all, the depolarisation field is not a bulk property of the ferroelectric, as it depends on the electrical boundary conditions. The authors attempt a boundary-independent definition of E_{dep} in Eq.(1), but this expression doesn't make too much sense to me, neither in an experimental or theoretical context. How would the authors calculate this quantity from first principles, for example? What would be its physical interpretation?

We do not claim that the depolarization field is boundary independent. In fact, the depolarization field is calculated from classical electrostatics within the boundary conditions of a thin film. The fictitious surface charge density is $\sigma = \mathbf{n} \cdot \mathbf{P}$ where \mathbf{P} is the polarization. The macroscopic electric field inside the thin film, the depolarization field E_{dep} , is equal to the field generated by such surface charge density, σ , and can be calculated by using the Gauss law, yielding $E_{dep} = -\frac{P}{\epsilon_0\epsilon_{ferro}}$, which appears as Eq. (1) in the manuscript.

First-principles calculations address the crystal field and electronic wavefunctions, and give the local field at an atomic scale; but that is beyond the scope of this work. The depolarization field that we study here is a macroscopic field arising from the aligned rigid dipoles^[4]. The existence of the depolarization field is a natural derivation from the framework of classic electrostatics dealing with a continuous medium.

A much more meaningful (and theoretically well defined) quantity to look at is the "intrinsic coercive field", which the authors discuss as well. This quantity can be calculated, either in a Landau or first-principles context, and related to measurable quantities (P_r and ϵ_{ferro}) via Eq.(10). I believe that the manuscript would much improve in clarity if the authors stated Eq.(10) upfront instead of their Eq.(1), possibly by giving some hints about its derivation and the assumptions that were made. Then say that they estimate $E_{c,int}$ via a measurement of ϵ_{ferro} and P_r . There is no need whatsoever to speak about the confusing and ill-defined E_{dep} , since $E_{c,int}$ is directly related to it.

Intrinsic switching

Intrinsic switching occurs at the intrinsic, or thermodynamic, coercive field $E_{int,c}$ where the Landau double potential well is destroyed. In the seminal work of Tagantsev *et al.*, a relation between the intrinsic coercive field, the polarization P_{LD} and the dielectric constant, has been theoretically derived as [Ref [5], Eq. 2.3.15, page 85]:

$$\frac{P_{LD}}{\epsilon_0 \epsilon_{LD} E_{int,c}} = 3\sqrt{3} \approx 5$$

This relation is reminiscent of our experimental derived relationship, *viz.* $P_r / \epsilon_0 \epsilon_{\text{ferro}} E_c \approx 15$. However, as will be explained below, the physical mechanisms are completely different, and cannot *a priori* be related.

Tagantsev *et al.* start with the mean-field treatment by Landau, where the Landau-Devonshire free energy, F , is expanded with an order parameter, the polarization P , as $F = -PE + \frac{1}{2}\alpha_0(T - T_0)P^2 + \frac{1}{4}\beta P^4$. Here P is the polarization, E is the electric field, α_0 ,

β and T_0 are Landau-Devonshire coefficients and T is the temperature. The stability condition, $dF/dP = 0$, leads to $E = \alpha_0(T - T_0)P + \beta P^3$. From this relation between E and P ,

the susceptibility is derived as $\chi_{LD} = \frac{1}{2\epsilon_0} \frac{\partial P}{\partial E} = \frac{1}{2\epsilon_0 \alpha_0 (T_0 - T)}$ and the saturated polarization P_{LD}

as $P_{LD} = \sqrt{\alpha_0(T_0 - T)/\beta}$ when taken $\left. \frac{\partial F}{\partial P} \right|_{E=0} = 0$. The magnitude of the intrinsic coercive

field is the extreme value of E , as $\partial E / \partial P = 0$, and reads $E_{int,c} = \frac{2}{3\sqrt{3}} \left(\frac{\alpha_0^3}{\beta} \right)^{1/2} (T_0 - T)^{3/2}$. By

eliminating the Landau-Devonshire coefficients, the relation $\frac{P_{LD}}{\epsilon_0 \epsilon_{LD} E_{int,c}} = 3\sqrt{3} \approx 5$ is obtained. Here we have taken the dielectric constant as $\epsilon_{LD} = \chi_{LD} + 1 \approx \chi_{LD}$.

Intrinsic switching is a mean-field process, which dictates collective behavior of the dipoles within the ferroelectric material. This happens when all dipoles are in a homogeneous environment; a necessary condition is that there are no pinning defects, which is practically unrealistic. Nevertheless, at very high electric field, the electrostatic energy dominates, thus influences from defects may become trivial: the pinning force by the defects is overwhelmed by the applied high field; the total electric field that the dipoles feel, either in defect-free regions or in vicinity of pinning defects, are about the same, and the system is effectively a homogeneous system. In that case, even a practical ferroelectric material with a considerable number of pinning defects may favour undergoing intrinsic switching.

Experimentally approaching the intrinsic switching has being a great challenge. To the best of our knowledge, experimental substantiation of intrinsic switching has only been reported in ultrathin films of BaTiO₃ and P(VDF-TrFE).

Ultrathin BaTiO₃(001) films were grown on a Pt(001)/MgO(001) substrate by laser ablation^[6]. The coercive field was extracted from AFM measurements and reported as a function of layer thickness^[7]. The coercive field increases with decreasing layer thickness. However, for thicknesses below about 10 nm the coercive field is constant, about 100 MV/m, therefore claimed to be the intrinsic coercive field. The depolarization field $P_{LD}/\epsilon_0 \epsilon_r E_{int,c}$ was calculated by these authors to be about 200 MV/m^[8] using a value of the dielectric constant of a single crystal BaTiO₃ of 150. This yields a value for $P_{LD}/\epsilon_0 \epsilon_{LD} E_{int,c} \sim 2$, in fair agreement with the theoretical value derived by Tagantsev *et al.* of about 5. Although this might suggest that $E_{dep} \sim E_{int}$, we note that BaTiO₃ is not a good model system as the dielectric constant varies over many orders of magnitude depending on chemical composition and microstructure. The dielectric constant of thin BaTiO₃ films also strongly depends on the film thickness^[9]. To unambiguously correlate the depolarization and activation field with the intrinsic coercive field, the comprising physical constants of the same device should be measured, which so far have not been reported.

P(VDF-TrFE) is an ideal model system as the dielectric constant hardly depends on microstructure and film thickness, and the complete data set on depolarization-, activation- and intrinsic coercive field is available. Intrinsic switching has been reported in Langmuir-Blodgett films of P(VDF-TrFE)^[10]. The ferroelectric films with thickness between 1 nm and 10 nm are thin enough to inhibit nucleation. The intrinsic coercive field is independent of layer thickness and has been presented as a function of temperature. At ambient temperature $E_{int,c}$ is measured to be about 600 MV/m. Taking the measured polarization as 0.1 C/m² and a dielectric constant of 10 then leads to a value for the depolarization field, $P_{LD}/\epsilon_0 \epsilon_r E_{int,c}$, of

about 1100 MV/m. The ratio $\frac{P_{LD}}{\epsilon_0 \epsilon_{LD} E_{int,c}} \approx 2$ is in good agreement with the theoretical value derived by Tagantsev *et al.* of about 5.

Previously the experimentally extracted activation field of P(VDF-TrFE) as a function of temperature has been reported^[11,12]. At ambient temperature a value of about 1000 MV/m was determined. This means that the activation field, depolarization field and intrinsic coercive field are of the same order of magnitude. Within a factor of two we arrive at $E_{dep} \sim E_{act} \sim E_{int,c}$.

However, we realize that this equality can be artificial as the underlying physical mechanisms for intrinsic and extrinsic switching are completely different. Current experiments, in our work as well as in literature, are insufficient to draw a solid conclusion about the relation with the intrinsic coercive field. Significantly more experiments are needed. For this reason, we have restricted our discussion to the activation field and the depolarization field.

The second quantity the authors look at is the quasi-static coercive field. This is a known quantity and techniques for measuring it well established. The third quantity is the activation field, E_{act} . I guess that this is the quantity that is hardest to measure (via depolarization dynamics), and the real conceptual/technical value of this work. The authors should state upfront that the largest part of the experimental effort, and the most remarkable novelty of this work, is the accurate measurement of E_{act} . Of course, the authors do not measure directly E_{act} , but only indirectly via some models and assumptions -- these should be clearly stated (i.e. what do the authors measure and what formula do they use to estimate E_{act} and why?)

Then, after the authors have these three quantities measured and the techniques for doing that well detailed, only then they should proceed at discussing their relationship, and at drawing some reasonable conclusions about the physical mechanism that is involved in switching. Right now all this material (discussions, speculations, etc.) is mixed up and I had to read the manuscript several times to figure out what is being done and how.

We are delighted at the comment of the reviewer regarding the “remarkable novelty of this work”. Thanks to their comments, we have restructured the discussion of the theoretical models used. In the revised manuscript, both the experimental extraction and the theoretical origin of E_{act} are stated in great detail, in order to derive the universal relation between activation and depolarization field.

Relation between activation field, E_{act} , and coercive field, E_c

Dipolar reversal at domain walls of ferroelectric materials leads to domain-wall motion, which is typically described by a creep velocity^[13]. The reciprocal domain-wall velocity is proportional to the switching time^[14], which follows the empirical Merz law^[15]:

$$t_0 = t_\infty \exp\left(\frac{E_{act}}{E}\right)$$

where E_{act} is the temperature dependent ‘‘activation field’’^[16] and t_∞ is the switching time at infinite applied electric field. Merz’s law is observed in many ferroelectric systems ranging from single crystals through bulk ceramics^[17], and thin films^[14,18], to organic-ferroelectric composites^[19]. We note that Tybel *et al.*^[13] were the first to point out that Merz’s law is a special case of domain-wall motion in generic creep systems, describing propagation of elastic objects driven by an external force in the presence of a pinning potential, such as domains in ferroelectric^[14] and magnetic materials^[20] and vortices in type-II superconductors^[21].

A few papers have addressed the relation between activation field and coercive field. A phenomenological expression has been suggested^[22]:

$$E_c \approx \frac{E_{act}}{\ln(\gamma E_c) - \ln(16f)}$$

where γ may be regarded as the displacement velocity of domains per volt. and γE_c has a large value and can be approximated as a constant. The coercive field then depends linearly on activation field.

Here we use P(VDF-TrFE) to experimentally derive the relation between activation field, coercive field and polarization. We emphasize that P(VDF-TrFE) is a unique model system. Contrary to most inorganic ferroelectric materials, the switching time for P(VDF-TrFE) can be measured over a wide range of electric field and temperature. We have reported the details of the measurements and data extraction in Ref.[11], and we here reproduce it for convenience from *S.I.* section 5. The extracted activation field as a function of the quasi-static coercive field is reproduced in Fig. 1.

Fig. 1| Activation field as a function of coercive field extracted from switching measurements on P(VDF-TrFE). The red line is a linear fit through the origin, resulting in a slope of ~ 15 .

Fig. 1 shows that the activation field linearly increases with the value of the coercive field: $E_{act}/E_c \approx 15$. A similar relation can be derived from the experimental data reported in Ref. [12]. The relation between and activation and coercive field has also theoretically been

addressed^[1,2,3]. The classical theory supporting Merz's law was developed by Miller and Weinreich^[1], derived for 180° domain-wall motion in BaTiO₃. The activation field is not an electric field, but a parameter related to the domain wall energy. The rate determining process is the nucleation of steps along the domain wall. The nucleation energy, ΔU^* , is determined by the competition between the cost of creating additional domain walls of the nucleus and the gain of the alignment with the applied electric field:

$$\Delta U^* = \frac{c\sigma_{dw}^2}{P_{sat}E_c}$$

where c is the width of the domain wall, σ_{dw} is the domain-wall energy and E_c is the coercive field. Subsequently, Miller and Weinreich arrived at an expression for the activation field that has been reformulated as^[12]:

$$E_{act} = \frac{c\sigma_{dw}^2}{P_{sat}k_B T}$$

By eliminating $c\sigma_{dw}^2$ we arrive at a relation between E_{act} and E_c as:

$$\frac{E_{act}}{E_c} = \Delta U^* / k_B T$$

A range of values for ΔU^* has been reported^[12], e.g. $10 k_B T$ for PZT^[23], $15 k_B T$ for PVDF Langmuir Blodgett films^[24], $29 k_B T$ for thin films of P(VDF-TrFE) and $40 k_B T$ for BaTiO₃^[25], leading to a value for the ratio of E_{act} over E_c in the range of 10 to 40.

We note that Miller and Weinreich suggested that the critical nucleus is an atomically thin triangular plate with a large aspect ratio, which then expands laterally on the same atomic plane. However, it is well-established that the Miller–Weinreich theory overestimates the activation field by an order of magnitude^[2]. Molecular dynamics simulations for 180° domain walls in defect-free PbTiO₃ did reveal not a triangular but a square critical nucleus with diffusive and beveled interfaces that substantially reduces the nucleation barrier and hence leads to much lower activation fields for domain-wall motion. However, here we are not calculating the absolute value of the activation field, but we only consider the ratio of E_{act} over E_c using experimentally extracted values of the nucleation energy. The proportionality constant is comparable to the one experimentally derived for P(VDF-TrFE), and agrees with molecular dynamics simulations.

Relation between activation field, E_{act} and depolarization field, E_{dep}

The values for E_{act}/E_c and $P_r/\epsilon_0\epsilon_{ferro}E_c$ are comparable. Therefore, we suggest that E_{act} is comparable to $P_r/\epsilon_0\epsilon_{ferro}$. As $P_r/\epsilon_0\epsilon_{ferro}$ is by definition equal to the depolarization field, E_{dep} , this then suggests that:

$$E_{act} \sim E_{dep}$$

This relation between values of activation- and depolarization field can be expected based on the origin of the activation field. At low electric field, the domain-wall motion is thermally activated and described by creep. The domains are pinned and the rate limiting step in domain growth is nucleation. With increasing electric field, the domain undergoes a pinning/depinning transition. Switching is still thermally activated but there is an increasing contribution of the electrostatic energy. At even higher fields, corresponding to the activation field, the nucleation barrier approaches zero; the electrostatic energy dominates. Domain wall motion is growth dominated only. The domain wall motion is then in the flow regime; the switching time depends linearly on electric field and does not depend on temperature^[3]. In this scenario the activation field corresponds to the transition threshold from creep to flow.

Here we consider pinning sites as dipoles with a fixed polarity. Dipoles in the vicinity tend to align in parallel to the polarity of the pinning site. We propose that depinning of the domain wall requires switching of these polarized regions. The average electric field within these polarized regions is the depolarization field, which has to be overcome in order to move the domain walls. Hence the depolarization field is the onset of domain-wall flow and, thus, similar to the activation field, $E_{dep} \sim E_{act}$.

After the major restructuring that I suggest above, I guess this could be a nice paper. I have a question on the physics, actually. The intrinsic coercive field assumes a monodomain switching passing through the saddle point of the Landau potential. This might be a well justified scenario in uniaxial ferroelectrics, but how about PZT? I guess that PZT might find some alternative paths, with substantially smaller barriers, by rotating the polarization vector, and therefore avoiding the very high-energy saddle point. Did the authors find a systematically smaller value of their universal constant in uniaxial ferroelectrics, by any chance?

In multi-axial ferroelectrics, the order parameter is the polarization vector, \mathbf{P} , and the Landau-Devonshire coefficients are tensors. The anisotropy and the coupling to mechanical stress modify the free energy. This can give rise to multiple energetic minima with distinctive polarization vectors or domain textures, as can be inferred from numerous studies over the local polarization using scanning probe microscope and supported by phase-field simulations.

However, a ferroelectric thin-film capacitor consists of numerous polycrystallites or grains with different size and orientation. In these systems, despite the existence of various domain textures in each crystallite or grain, the electrically measured macroscopic polarization is binary, as a result of ensemble average. Properties such as the activation field and coercive field are obtained from electric measurements on thin-film capacitors consisting of multiple crystallites or grains. Therefore, the universal constant we discussed in the manuscript is not sensitive to the polarization axis of single domains.

The suggestion of the reviewer is more than worthwhile. However, to investigate differences between uniaxial and multi-axial ferroelectrics, we need a capacitor made of a single-crystal thin-film without any defects, such as LiNbO₃ or LiTaO₃. At present, for us, these capacitors are not accessible.

Comments of Reviewer #2 and answers by the authors:

We thank the reviewer for critically reading the manuscript and for their constructive comments. All the points raised have been addressed in the revised manuscript and are discussed below. As a consequence, the quality of the manuscript has increased significantly.

In their paper “Depolarization of multidomain ferroelectric materials” Katsouras et al. study domain wall depinning and depolarization in a dynamic regime. They especially infer a single constant to describe the boundary between a stable ferroelectric state and a region where depolarization governs. The effect of depolarization and how to engineer and understand stable ferroelectric states is important and has been a field of large research for more than 40 years. In this work the authors study these effects in an electric circuit setting including series capacitances, whilst traditionally one has tried to investigate in proper short circuit conditions. Taken together, the authors have presented a solid study, however I feel that the presented data and analysis is better suited for a more specialized journal when the points above are addressed.

I do have some concerns regarding the paper that should be clarified to better convey the message to the readers:

1. The authors perform a lump circuit electrostatic analysis. In such an analysis they do not take into concern the effect of the finite screening of the electrode. It has been shown theoretically that due to a finite screening length of the electrodes the internal field will never be zero inside a ferroelectric, even under short circuit conditions. How does this affect the statements made in the paper of a regime with zero depolarization?

In the introduction we mentioned that incomplete screening of the bound polarization charge at the ferroelectric-electrode interface leads to a depolarization field^[26,27,28]. There are two reasons for incomplete screening. Firstly, the compensating charges in the electrode form a layer of finite thickness due to Thomas-Fermi screening length. Secondly, the polarization cannot drop abruptly when going from the ferroelectric to the metal, the so-called Kretschmer-Binder effect.

The effect of this depolarization field will become larger as the thickness of the ferroelectric decreases. The depolarization field is especially important in ultrathin films in the order of 10 nm, where it determines the critical thickness and domain structure.

For thick films the depolarization field from incomplete screening can be disregarded. Only when the ferroelectric is a perfect insulator, the incomplete screening leads to a finite depolarization field inside the ferroelectric material. However, due to the large film thickness this internal electric field is much smaller than the coercive field. Secondly, and more importantly, ferroelectric materials are not perfect insulators; tangent delta is finite and not zero. The uncompensated charges by the ferroelectric-electrode interface are neutralized by charge carriers in the ferroelectric material. Consequently, inside a thick film ferroelectric

capacitor under short circuit conditions the internal electric field is zero. This statement is supported by the measured polarization of our samples, which is thickness independent. Furthermore, in ultrathin 15 nm BaTiO₃ films sandwiched between SrRuO₃ electrodes already 80 % of the remanent polarization is retained^[29]. Finally, the remanent polarization of ultra-thin PbTiO₃ films saturates above 20 nm^[30]. Therefore, in our electrostatic analysis the depolarization field due to incomplete screening in the electrodes can be disregarded.

The comments of the reviewer are highly appreciated as they pointed out that the introduction was not clear. In the updated manuscript we explicitly state that depolarization due to screening effects of the electrodes can be disregarded as the analysis is done for thick films.

2. In the experiments the authors use the value of the series capacitance as control parameter, whilst the ferroelectric capacitance is constant. When changing the series capacitor, the system will also exhibit a change of effective RC time constant, and a saturation would be expected for small series capacitances, that is in the same region where the authors claim a thermodynamic stable state? Have the authors tried to perform similar measurements with C series constant and effectively changing the ferroelectric capacitance systematically?

We changed the capacitance of the P(VDF-TrFE) capacitor by an order of magnitude, and repeated the measurements. Similar data were obtained, *i.e.* the apparent polarization scales uniquely with the ratio of C_{ferro}/C_{ser} . Similarly, we showed in the manuscript that the suppression of the polarization as a function of C_{ferro}/C_{ser} is the same for P(VDF-TrFE) and PZT capacitors, while the capacitances are hugely different. Hence the presented polarization as a function of C_{ferro}/C_{ser} is not affected by an effective RC time constant.

3. The authors claim that the constant $P/(\epsilon_0 \epsilon_f E_c)$ is a universal constant, however in the paper they show data where the constant varies within a factor 3-4. Moreover, they claim that the constant is also valid close to a phase transition (table 2), this is not clear from the theoretical part of the paper why it should be valid close to a phase transition? If a universal constant one would assume it to be the same for a given material, and maybe deviate close to phase transitions?

We took the extracted values for $P_r/\epsilon_0 \epsilon_{ferro} E_c$ as presented in Table 1, and calculated the standard deviation, yielding a value 15 ± 4 . As we wrote in the manuscript “among different ferroelectric materials the values of coercive field and dielectric constant vary by orders of magnitude. Remarkably, for the wide variety of ferroelectric materials considered here, a comparable value for $P_r/\epsilon_0 \epsilon_{ferro} E_c$ is obtained, suggesting that this constant is universal”. Therefore, we thank the reviewer for the comment, and to prevent any confusion we added the standard deviation in the revised manuscript.

The exact behaviour of $P_r/\epsilon_0 \epsilon_{ferro} E_c$ at/above the ferroelectric/paraelectric phase transition temperature is unknown. Our analysis holds below the Curie temperature where we,

experimentally, show for P(VDF-TrFE) that $P_r/\epsilon_0\epsilon_{ferro}E_c$ is temperature independent, supporting that the constant is universal (Fig.2 below). To obtain the temperature dependence, we have measured the polarization, dielectric constant and coercive field of P(VDF-TrFE) ferroelectric-only capacitors at temperatures between 213 K and 333 K, as explained in detail in section 5 of the SI.

Fig. 2| Extracted value of $P_r/\epsilon_0\epsilon_{ferro}E_c$ as a function of temperature for P(VDF-TrFE)

Comments of Reviewer #3 and answers by the authors:

We thank the reviewer for the positive comments. We highly appreciate that “The authors use an ingenious method to apply an electric field through the depolarization mechanism and use it to derive a universal relationship between the remanent polarization, dielectric constant and coercive field of ferroelectric materials”.

We fully agree that there were ambiguities in the submitted paper. In the revised manuscript we have addressed all the points raised. Thanks to the comments of the reviewer the discussion part in the revised manuscript has improved significantly. We hope that considering the changes made, the manuscript is now suitable for publication in *Nature Communications*.

To facilitate the reviewer, our response is structured as follows. First all the comments are copied. Subsequently, although we agree with the reviewer that the physical mechanism of intrinsic switching is different from the nucleation and growth studied here for extrinsic switching, we include a brief discussion on intrinsic switching for clarity. We then focus on extrinsic switching by domain wall motion. We experimentally derive a linear relation between activation field and coercive field. The relation is supported by the classical theory of Miller and Weinreich^[1], and by molecular dynamic calculations^[2,3]. Then, by using the experimentally extracted universal value of $P_r/\epsilon_0\epsilon_{ferro}E_c$ of about 15, we can connect the activation field with the depolarization field. We show that the depolarization field is the onset of domain-wall flow and similar to the activation field, $E_{dep} \sim E_{act}$.

Finally, we will address the screening length of the electrode and show that it does not affect our experimental data or analysis.

Comments of reviewer 3

The work described in the paper is quite interesting. The authors use an ingenious method to apply an electric field through the depolarization mechanism and use it to derive a universal relationship between the remanent polarization, dielectric constant and coercive field of ferroelectric materials. While I think the experimental results are quite important and general, I have some questions about their interpretation by the authors. The authors argue that the activation field of Merz's law is the depolarization field and ascribe this to depinning from defects. They use the original model proposed by Miller and Weinreich to explain the origin of the activation field and also try to link their result for the coercive field being proportional to P/ϵ to the intrinsic coercive field as described by Tagantsev et al. I think this is either explained in a confusing way or a mistake.

First, the intrinsic coercive field derived in the book by Tagantsev et al. is the field at which one of the double wells in the FE double well potential is destroyed. At this field, the switching process is intrinsic, that is to say the entire material will switch simultaneously on the ps timescale. This is clearly not relevant to the extrinsic, nucleation-and-growth driven

domain wall motion process that proceeds on much slower timescales precisely because the different domains are located in the different sides of the double well potential energy surface.

Second, the extrinsic DW motion process is thought to be controlled by the nucleation energy which is determined by the competition between the cost of creating additional domain walls of the nucleus and the gain of the alignment with the applied E field, as described first by the Miller and Weinreich model cited by the authors. Thus, the coercive field is (roughly) determined by the ratio of the square of the domain wall energy and polarization (σ^2/P). The authors find that the coercive field is proportional to P/ϵ . Perhaps the σ^2/P ratio of the Miller-Weinreich model is proportional to the P/ϵ ratio derived by Tagantsev for the intrinsic coercive field and experimentally by the authors for the measured coercive field, but it is not clear why these two quantities (σ^2/P) and (P/ϵ) should be linearly related and whether for example they have a similar temperature dependence. The authors should demonstrate the equivalence between these two formulas for E_c . Alternatively, if these two formulas are not equivalent, this would indicate that the Miller-Weinreich model is inapplicable for the switching studied in this work. The finding that E_c does not follow the standard nucleation model would be quite unusual and would lead to the question of what mechanism governs the switching observed by the authors and what relationship this mechanism has to domain wall motion.

Third, the activation field in Merz's law is not really a field in the same sense as an applied electric field or the depolarizing field. The activation field is merely a bookkeeping device that collects all of the constants in the Miller-Weinreich formula or alternatively a parameter that sets the dependence of the domain wall velocity on the applied field. Therefore, identifying the activation field with the actual existing electric field in the material (the depolarization field) is confusing. The fact that the coercive field is about ten times smaller than the activation field means that thermal energy that is ten times smaller than the barrier height is sufficient to provide the thermal fluctuations for switching to occur. It is not clear whether or how this is related to the nature of the field applied to drive the nucleation process. Thus, the statement that $E_{\text{depol}} \sim E_{\text{act}}$ is in not very physically meaningful, because this merely says that E_{depol} has the same magnitude as $E_{\text{activation}}$ but in no way implies that E_{depol} is the field that governs the switching in all FE materials where $E_{\text{activation}}$ is observed to be ten times larger than E_c .

Detailed answer to the questions raised.

Intrinsic switching

Intrinsic switching occurs at the intrinsic, or thermodynamic, coercive field $E_{\text{int},c}$ where the Landau double potential well is destroyed. In the seminal work of Tagantsev *et al.*, a relation between the intrinsic coercive field, the polarization P_{LD} and the dielectric constant, has been theoretically derived as [Ref [5], Eq. 2.3.15, page 85]:

$$\frac{P_{LD}}{\epsilon_0 \epsilon_{LD} E_{int,c}} = 3\sqrt{3} \approx 5$$

This relation is reminiscent of our experimental derived relationship, *viz.* $P_r/\epsilon_0 \epsilon_{ferro} E_c \approx 15$. However, as will be explained below, the physical mechanisms are completely different, and cannot *a priori* be related.

Tagantsev *et al.* start with the mean-field treatment by Landau, where the Landau-Devonshire free energy, F , is expanded with an order parameter, the polarization P , as $F = -PE + \frac{1}{2}\alpha_0(T - T_0)P^2 + \frac{1}{4}\beta P^4$. Here P is the polarization, E is the electric field, α_0 , β and T_0 are Landau-Devonshire coefficients and T is the temperature. The stability condition, $dF/dP = 0$, leads to $E = \alpha_0(T - T_0)P + \beta P^3$. From this relation between E and P , the susceptibility is derived as $\chi_{LD} = \frac{1}{2\epsilon_0} \frac{\partial P}{\partial E} = \frac{1}{2\epsilon_0 \alpha_0 (T_0 - T)}$ and the saturated polarization P_{LD} as $P_{LD} = \sqrt{\alpha_0(T_0 - T)/\beta}$ when taken $\left. \frac{\partial F}{\partial P} \right|_{E=0} = 0$. The magnitude of the intrinsic coercive

field is the extreme value of E , as $\partial E/\partial P = 0$, and reads $E_{int,c} = \frac{2}{3\sqrt{3}} \left(\frac{\alpha_0^3}{\beta} \right)^{1/2} (T_0 - T)^{3/2}$. By eliminating the Landau-Devonshire coefficients, the relation $\frac{P_{LD}}{\epsilon_0 \epsilon_{LD} E_{int,c}} = 3\sqrt{3} \approx 5$ is obtained. Here we have taken the dielectric constant as $\epsilon_{LD} = \chi_{LD} + 1 \approx \chi_{LD}$.

Intrinsic switching is a mean-field process, which dictates collective behavior of the dipoles within the ferroelectric material. This happens when all dipoles are in a homogeneous environment; a necessary condition is that there are no pinning defects, which is practically unrealistic. Nevertheless, at very high electric field, the electrostatic energy dominates, thus influences from defects may become trivial: the pinning force by the defects is overwhelmed by the applied high field; the total electric field that the dipoles feel, either in defect-free regions or in vicinity of pinning defects, are about the same, and the system is effectively a homogeneous system. In that case, even a practical ferroelectric material with a considerable number of pinning defects may favour undergoing intrinsic switching.

Experimentally approaching the intrinsic switching has being a great challenge. To the best of our knowledge, experimental substantiation of intrinsic switching has only been reported in ultrathin films of BaTiO₃ and P(VDF-TrFE).

Ultrathin BaTiO₃(001) films were grown on a Pt(001)/MgO(001) substrate by laser ablation^[6]. The coercive field was extracted from AFM measurements and reported as a function of layer thickness^[7]. The coercive field increases with decreasing layer thickness. However, for thicknesses below about 10 nm the coercive field is constant, about 100 MV/m, therefore

claimed to be the intrinsic coercive field. The depolarization field $P_{LD}/\epsilon_0\epsilon_r E_{int,c}$ was calculated by these authors to be about 200 MV/m^[8] using a value of the dielectric constant of a single crystal BaTiO₃ of 150. This yields a value for $P_{LD}/\epsilon_0\epsilon_{LD}E_{int,c} \sim 2$, in fair agreement with the theoretical value derived by Tagantsev *et al.* of about 5. Although this might suggest that $E_{dep} \sim E_{int}$, we note that BaTiO₃ is not a good model system as the dielectric constant varies over many orders of magnitude depending on chemical composition and microstructure. The dielectric constant of thin BaTiO₃ films also strongly depends on the film thickness^[9]. To unambiguously correlate the depolarization and activation field with the intrinsic coercive field, the comprising physical constants of the same device should be measured, which so far have not been reported.

P(VDF-TrFE) is an ideal model system as the dielectric constant hardly depends on microstructure and film thickness, and the complete data set on depolarization-, activation- and intrinsic coercive field is available. Intrinsic switching has been reported in Langmuir-Blodgett films of P(VDF-TrFE)^[10]. The ferroelectric films with thickness between 1 and 10 nm are thin enough to inhibit nucleation. The intrinsic coercive field is independent of layer thickness and has been presented as a function of temperature. At ambient temperature $E_{int,c}$ is measured to be about 600 MV/m. Taking the measured polarization as 0.1 C/m² and a dielectric constant of 10 then leads to a value for the depolarization field, $P_{LD}/\epsilon_0\epsilon_r E_{int,c}$, of

about 1100 MV/m. The ratio $\frac{P_{LD}}{\epsilon_0\epsilon_{LD}E_{int,c}} \approx 2$ is in good agreement with the theoretical value

derived by Tagantsev *et al.* of about 5.

Previously the experimentally extracted activation field of P(VDF-TrFE) as a function of temperature has been reported^[11,12]. At ambient temperature a value of about 1000 MV/m was determined. This means that the activation field, depolarization field and intrinsic coercive field are of the same order of magnitude. Within a factor of two we arrive at $E_{dep} \sim E_{act} \sim E_{int,c}$.

However, we realize that this equality can be artificial as the underlying physical mechanisms for intrinsic and extrinsic switching are completely different. Current experiments, in our work as well as in literature, are insufficient to draw a solid conclusion about the relation with the intrinsic coercive field. Significantly more experiments are needed. For this reason, we have restricted our discussion to the activation field and the depolarization field.

Relation between activation field, E_{act} , and coercive field, E_c

Dipolar reversal at domain walls of ferroelectric materials leads to domain-wall motion, which is typically described by a creep velocity^[13]. The reciprocal domain-wall velocity is proportional to the switching time^[14], which follows the empirical Merz law^[15]:

$$t_0 = t_\infty \exp\left(\frac{E_{act}}{E}\right)$$

where E_{act} is the temperature dependent “activation field”^[16] and t_∞ is the switching time at infinite applied electric field.

We note that we fully agree with the reviewer that E_{act} , as it appears in Merz’s law, is not a real electric field but a phenomenological parameter with units of electric field, which is related to the energy required to move the domain walls. Here we derive that the energy barrier to overcome for domain wall motion corresponds to an actual electric field, *i.e.* the depolarization field. Hence the depolarization field, as the onset of domain-wall flow, has a similar magnitude to the phenomenological activation field, $E_{dep} \sim E_{act}$.

Merz’s law is observed in many ferroelectric systems ranging from single crystals through bulk ceramics^[17], and thin films^[14,18], to organic-ferroelectric composites^[19]. We note that Tybel *et al.*^[13] were the first to point out that Merz’s law is a special case of domain-wall motion in generic creep systems, describing propagation of elastic objects driven by an external force in the presence of a pinning potential, such as domains in ferroelectric^[14] and magnetic materials^[20] and vortices in type-II superconductors^[21].

A few papers have addressed the relation between activation field and coercive field. A phenomenological expression has been suggested^[22]:

$$E_c \approx \frac{E_{act}}{\ln(\gamma E_c) - \ln(16f)}$$

where γ may be regarded as the displacement velocity of domains per volt and γE_c has a large value and can be approximated as a constant. The coercive field then depends linearly on activation field.

Here we use P(VDF-TrFE) to experimentally derive the relation between activation field, coercive field and polarization. We emphasize that P(VDF-TrFE) is a unique model system. Contrary to most inorganic ferroelectric materials, the switching time for P(VDF-TrFE) can be measured over a wide range of electric field and temperature. We have reported the details of the measurements and data extraction in Ref.[11], and we here reproduce it for convenience from *S.I.* section 5. The extracted activation field as a function of the quasi-static coercive field is reproduced in Fig. 3.

Fig. 3| Activation field as a function of coercive field extracted from switching measurements on P(VDF-TrFE). The red line is a linear fit through the origin, resulting in a slope of ~ 15 .

Fig. 3 shows that the activation field linearly increases with the value of the coercive field: $E_{act}/E_c \approx 15$. A similar relation can be derived from the experimental data reported in Ref. [12]. The relation between activation and coercive field has also theoretically been addressed. [1,2,3] The classic theory supporting Merz's law was developed by Miller and Weinreich^[1], derived for 180° domain-wall motion in BaTiO₃. The activation field is not an electric field, but a parameter related to the domain wall energy. The rate determining process is the nucleation of steps along the domain wall. The nucleation energy, ΔU^* , is determined by the competition between the cost of creating additional domain walls of the nucleus and the gain of the alignment with the applied electric field:

$$\Delta U^* = \frac{c\sigma_{dw}^2}{P_{sat}E_c}$$

where c is the width of the domain wall, σ_{dw} is the domain-wall energy and E_c is the coercive field. Subsequently, Miller and Weinreich arrived at an expression for the activation field that has been reformulated as^[12]:

$$E_{act} = \frac{c\sigma_{dw}^2}{P_{sat}k_B T}$$

By eliminating $c\sigma_{dw}^2$ we arrive at a relation between E_{act} and E_c as:

$$\frac{E_{act}}{E_c} = \Delta U^* / k_B T$$

A range of values for ΔU^* has been reported^[12], e.g. $10 k_B T$ for PZT^[23], $15 k_B T$ for PVDF Langmuir Blodgett films^[24], $29 k_B T$ for thin films of P(VDF-TrFE) and $40 k_B T$ for BaTiO₃^[25], leading to a value for the ratio of E_{act} over E_c in the range of 10 to 40.

We note that Miller and Weinreich suggested that the critical nucleus is an atomically thin triangular plate with a large aspect ratio, which then expands laterally on the same atomic plane. However, it is well-established that the Miller–Weinreich theory overestimates the

activation field by an order of magnitude^[2]. Molecular dynamics simulations for 180° domain walls in defect-free PbTiO₃ did reveal not a triangular but a square critical nucleus with diffusive and beveled interfaces that substantially reduces the nucleation barrier and hence leads to much lower activation fields for domain-wall motion. However, here we are not calculating the absolute value of the activation field, but we only consider the ratio of E_{act} over E_c using experimentally extracted values of the nucleation energy. The proportionality constant is comparable to the one experimentally derived for P(VDF-TrFE), and agrees with molecular dynamics simulations.

Relation between activation field, E_{act} and depolarization field, E_{dep}

The values for E_{act}/E_c and $P_r/\epsilon_0\epsilon_{ferro}E_c$ are comparable. Therefore, we suggest that E_{act} is comparable to $P_r/\epsilon_0\epsilon_{ferro}$. As $P_r/\epsilon_0\epsilon_{ferro}$ is by definition equal to the depolarization field, E_{dep} , this then suggests that:

$$E_{act} \sim E_{dep}$$

This relation between values of activation- and depolarization field can be expected based on the origin of the activation field. At low electric field, the domain-wall motion is thermally activated and described by creep. The domains are pinned and the rate limiting step in domain growth is nucleation. With increasing electric field, the domain undergoes a pinning/depinning transition. Switching is still thermally activated but there is an increasing contribution of the electrostatic energy. At even higher fields, corresponding to the activation field, the nucleation barrier approaches zero; the electrostatic energy dominates. Domain wall motion is growth dominated only. The domain wall motion is then in the flow regime; the switching time depends linearly on electric field and does not depend on temperature^[3]. In this scenario the activation field corresponds to the transition from creep to flow.

Here we consider pinning sites as dipoles with a fixed polarity. Dipoles in the vicinity tend to align in parallel to the polarity of the pinning site. We propose that depinning of the domain wall requires switching of these polarized regions. The average electric field within these polarized regions is the depolarization field, which has to be overcome in order to move the domain walls. Hence the depolarization field is the onset of domain-wall flow and, thus, similar to the activation field, $E_{dep} \sim E_{act}$.

Screening length of the electrodes.

Finally, the authors use the results of previous P_r , epsilon and E_c measurements to demonstrate the universality of the relationship between E_c and P/ϵ . If these data were taken from the measurements using the standard electrodes, would the authors approach for estimating E_{dep} be still valid, considering that in the presence of electrodes, E_{dep} depends on the screening length of the electrodes and on the thickness of the ferroelectric? The authors should explain this.

In the introduction we mentioned that incomplete screening of the bound polarization charge at the ferroelectric-electrode interface leads to a depolarization field^[26,27,28]. There are two reasons for incomplete screening. Firstly, the compensating charges in the electrode form a layer of finite thickness due to Thomas-Fermi screening length. Secondly, the polarization cannot drop abruptly when going from the ferroelectric to the metal, the so-called Kretschmer-Binder effect.

The effect of this depolarization field will become larger as the thickness of the ferroelectric decreases. The depolarization field is especially important in ultrathin films in the order of 10 nm, where it determines the critical thickness and domain structure.

For thick films the depolarization field from incomplete screening can be disregarded. Only when the ferroelectric is a perfect insulator, the incomplete screening leads to a finite depolarization field inside the ferroelectric material. However, due to the large film thickness this internal electric field is much smaller than the coercive field. Secondly, and more importantly, ferroelectric materials are not perfect insulators; tangent delta is finite and not zero. The uncompensated charges by the ferroelectric-electrode interface are neutralized by charge carriers in the ferroelectric material. Consequently, inside a thick film ferroelectric capacitor under short circuit conditions the internal electric field is zero. This statement is supported by the measured polarization of our samples, which is thickness independent. Furthermore, in ultrathin 15 nm BaTiO₃ films sandwiched between SrRuO₃ electrodes already 80 % of the remanent polarization is retained^[29]. Finally, the remanent polarization of ultra-thin PbTiO₃ films saturates above 20 nm^[30].

Therefore, in our electrostatic analysis the depolarization field due to incomplete screening in the electrodes can be disregarded.

We appreciate the comment of the reviewer, as it pointed out that the introduction was ambiguous. In the updated manuscript we explicitly state that depolarization due to screening effects of the electrodes can be disregarded as the analysis is done for thick films.

List of Changes

We have restructured and made major changes that significantly improved the manuscript. To keep the list of changes concise, we list below the parts of the manuscript that have been changed. For completeness, all changes have been highlighted in the resubmitted manuscript.

- We added a coauthor and the list of authors has been changed.
- For clarity and better readability, we have structured the manuscript in subsections.
- The last sentence of the abstract has been modified to reflect the changes in the main text.
- We have re-written the last part of the introduction, according to the comments of the reviewers regarding the screening length of the electrodes. Furthermore, we clarified the relation of our work to the seminal work of Tagantsev *et. al.*.
- The major part of the paper, the results, has hardly been changed, as the reviewers praised the ingenuity of the method to study depolarization.
- We have added an additional figure with experimental data. The temperature independence presented, shows that the $P_r/\epsilon_0\epsilon_{ferro}E_c$ parameter is really universal. This is supported by the statistical average of the $P_r/\epsilon_0\epsilon_{ferro}E_c$ parameter presented in Table 1.
- The discussion part is completely re-written, to clarify all ambiguities of the original manuscript, as suggested by the reviewers.
- The SI has been updated. Two sections on electrode screening length and intrinsic switching have been added, according to the suggestions of the reviewers.
- We have adapted the last sentence of the summary.
- All references have been updated.

References

- [1] Miller, R. C. & Weinreich, G. Mechanism for the sidewise motion of 180° domain walls in barium titanate. *Phys. Rev.* **117**, 1460-1466 (1960).
- [2] Shin, Y.-H., Grinberg, I., Chen, I.-W. & Rappe, A. M. Nucleation and growth mechanism of ferroelectric domain-wall motion. *Nature* **449**, 881-884 (2007).
- [3] Shi, L., Grinberg, I. & Rappe, A. M., Intrinsic ferroelectric switching from first principles. *Nature* **534**, 360-363 (2016).
- [4] C. Kittel, Introduction to Solid State Physics Ed. 8 (Wiley)
- [5] Tagantsev, A., Cross, L. & Fousek, J. Domains in Ferroic Crystals and Thin Films (Springer, 2010).
- [6] R. Gaynutdinov, M. Minnekaev, S. Mitko, A. Tolstikhina, A. Zenkevich, S. Ducharme, V. Fridkin, Polarization switching kinetics in ultrathin ferroelectric barium titanate film, *Physica B* **424** (2013) 8–12.
- [7] R. Gaynutdinov, M. Minnekaev, S. Mitko, A. Tolstikhina, A. Zenkevich, S. Ducharme, V. Fridkin, Scaling of the Coercive Field in Ferroelectrics at the Nanoscale, *JETP Letters*, November 2013, Volume 98, Issue 6, pp 339–341.
- [8] V. M. Fridkin, S. Ducharme, General features of the intrinsic ferroelectric coercive field, *Physics of the Solid State*, July 2001, Volume 43, Issue 7, pp 1320–1324
- [9] Y. Yano, K. Iijima, Y. Daitoh, T. Terashima, and Y. Bando, Epitaxial growth and dielectric properties of BaTiO₃ films on Pt electrodes by reactive evaporation, *Journal of Applied Physics* **76**, 7833 (1994)
- [10] Ducharme, S., Fridkin, V.M., Bune, A.V., Palto, S.P., Blinov, L.M., Petukhova, N.N., Yudin, S.G., *Phys. Rev. Lett.* **84**, 175 (2000).
- [11] Zhao, D., Katsouras, I., Asadi, K., Blom, P. W. M. & de Leeuw, D. M. Switching dynamics in ferroelectric P(VDF-TrFE) thin films. *Phys. Rev. B* **92**, 214115 (2015).
- [12] W. Hu, D. Juo, L. You, J. Wang, Y. Chen, Y. Chu, and T. Wu, Universal Ferroelectric Switching Dynamics of Vinylidene Fluoride-trifluoroethylene Copolymer Films, *Sci. Rep.* **4**, 4772 (2014).
- [13] Tybell, T., Paruch, P., Giamarchi, T. & Triscone, J., Domain Wall Creep in Epitaxial Ferroelectric Pb(Zr_{0.2}Ti_{0.8})O₃ Thin Films. *Phys. Rev. Lett.* **89**, 9 (2002).

-
- [14] Jo, J. et al., Nonlinear Dynamics of Domain-Wall Propagation in Epitaxial Ferroelectric Thin Films. *Phys. Rev. Lett.* 102, 045701 (2009).
- [15] Merz, W. J. Domain formation and domain wall motions in ferroelectric BaTiO₃ single crystals. *Phys. Rev.* 95, 690-698 (1954).
- [16] Shin, Y.-H., Grinberg, I., Chen, I.-W. & Rappe, A. M. Nucleation and growth mechanism of ferroelectric domain-wall motion. *Nature* 449, 881-884 (2007).
- [17] Zhukov, S. et al. Dynamics of polarization reversal in virgin and fatigued ferroelectric ceramics by inhomogeneous field mechanism. *Phys. Rev. B* 82, 014109 (2010).
- [18] Gruverman, A., Wu, D. & Scott, J. F. Piezoresponse force microscopy studies of switching behavior of ferroelectric capacitors on a 100-ns time scale. *Phys. Rev. Lett.* 100, 097601 (2008).
- [19] Nautiyal, A. et al. Polarization switching properties of spray deposited CsNO₃: PVA composite films. *Appl. Phys. A* 99, 941-946 (2010).
- [20] Lemerle, S. et al. Domain wall creep in an Ising ultrathin magnetic film. *Phys. Rev. Lett.* 80, 849-852 (1998).
- [21] Blatter, G., Feigel'man, M. V., Geshkenbein, V. B., Larkin, A. I. & Vinokur, V. M. Vortices in high-temperature superconductors. *Rev. Mod. Phys.* 66, 1125-1388 (1994).
- [22] C. Pulvari and W. Kuebler, Phenomenological Theory of Polarization Reversal in BaTiO₃ Single Crystals, *J. Appl. Phys.* 29, 1315-1321 (1958).
- [23] Yang, S. M. et al. Ac dynamics of ferroelectric domains from an investigation of the frequency dependence of hysteresis loops. *Phys. Rev. B* 82, 174125 (2010).
- [24] Sharma, P., Reece, T. J., Ducharme, S. & Gruverman, A. High-Resolution Studies of Domain Switching Behavior in Nanostructured Ferroelectric Polymers. *Nano Letters* 11, 1970–1975 (2011).
- [25] Jo, J. Y. et al. Polarization switching dynamics governed by thermodynamic nucleation process in ultrathin ferroelectric films. *Phys. Rev. Lett.* 97, 247602 (2006).
- [26] Mehta, R. R., Silverman, B. D. & Jacobs, J. T. Depolarization fields in thin ferroelectric films. *J. Appl. Phys.* 44, 3379-3385 (1973).
- [27] Kim, D. J. et al. Polarization relaxation induced by a depolarization field in ultrathin ferroelectric BaTiO₃ capacitors. *Phys. Rev. Lett.* 95, 237602 (2005).

[28] Gerra, G., Tagantsev, A. K., Setter, N. & Parlinski, K. Ionic polarizability of conductive metal oxides and critical thickness for ferroelectricity in BaTiO₃. Phys. Rev. Lett. 96, 107603 (2006).

[29] Junquera, J., Ghosez, P., Critical thickness for ferroelectricity in perovskite ultra-thin films, Nature 422, 506 (2003).

[30] Lichtensteiger, C., Triscone, J., Junquera J., Ghosez, P., Phys. Rev. Lett. 94, 047603 (2005).

REVIEWERS' COMMENTS:

Reviewer #1 (Remarks to the Author):

I believe that the authors have done a serious and diligent job at addressing my comments and criticisms. I have no further objections to publication.

Reviewer #2 (Remarks to the Author):

I find that the authors have addressed my previous concerns in a good manner and can now recommend the paper for publication.

Reviewer #3 (Remarks to the Author):

I think this paper has been much improved by the authors' revisions. As I have said before, I think the authors' method and results are quite interesting and are suitable in principle for publication in Nature Communications. However, I still do not understand the authors' explanation for how the depolarizing field is related to the coercive field and the activation field. The authors describe this relationship in words and state that the depolarizing field is the field at the crossover between the creep and flow regimes and is related to the pinning sites in the material. I could not follow the explanation for this in the paper. Therefore, the authors should provide a mathematical description of a model for switching that would show this relationship more clearly. Alternatively, the authors can remove this claim from the paper and just note the similarity between E_{depol} and E_{act} as they now do for E_{act} and the intrinsic coercive fields. Either option is fine, but as written this part of the paper is not clear and may be wrong.